# Estimating the global burden of viable *Mycobacterium tuberculosis* infection: A mathematical modelling study

Alvaro Schwalb[1,2,3]*, Peter J. Dodd[4], Hannah M. Rickman[5,6], César A. Ugarte-Gil[7], Katherine C. Horton[1,2], Rein M. G. J. Houben[1,2]

**1** TB Modelling Group, TB Centre, London School of Hygiene and Tropical Medicine, London, United Kingdom, **2** Department of Infectious Disease Epidemiology, London School of Hygiene and Tropical Medicine, London, United Kingdom, **3** Instituto de Medicina Tropical Alexander von Humboldt, Universidad Peruana Cayetano Heredia, Lima, Peru, **4** School of Health and Related Research, University of Sheffield, Sheffield, United Kingdom, **5** Clinical Research Department, London School of Hygiene and Tropical Medicine, London, United Kingdom, **6** Malawi Liverpool Wellcome Programme, Blantyre, Malawi, **7** School of Public and Population Health, University of Texas Medical Branch, Galveston, Texas, United States of America

\* alvaro.schwalb@lshtm.ac.uk

## Abstract

### Background

Estimating the proportion of individuals currently infected with *Mycobacterium tuberculosis* (*Mtb*) is key for informing global health policies. Although a substantial portion of the global population exhibit tuberculous immunoreactivity, not all have a viable *Mtb* infection. Moreover, individuals with recent infections are at a higher risk of developing tuberculosis (TB). Here, we present estimates of the global burden of viable *Mtb* infection, using new insights into the natural history of TB.

### Methods and findings

We constructed country-specific trends in annual risk of infection considering estimates of TB burden, immunoreactivity reversion, and age-specific mixing. We applied these trends to a deterministic mathematical model incorporating reinfection and self-clearance to estimate recent (within 2 years) and total viable *Mtb* infections. Empirical data on self-clearance are limited, so rates were informed by modelling estimates. In 2022, we estimated that 133.7 million people (95% uncertainty interval [UI]: 104.0, 171.1) had a recent *Mtb* infection, representing 1.7% (95% UI: 1.3, 2.2) of the global population. In total, 288.9 million people (95% UI: 242.2, 342.7)—or 3.7% (95% UI: 3.1, 4.3) globally—were estimated to harbour a viable *Mtb* infection. Among those recently infected, 12.0% (95% UI: 11.4, 12.7) were children under 15 years of age. Most recent infections were found in the World Health Organization regions of South-East Asia (49.0%; 95% UI: 37.2, 62.4), the Western Pacific (19.7%; 95% UI: 12.6, 30.5), and Africa (17.9%; 95% UI: 12.9, 24.1). India, Indonesia, and China

**Data availability statement:** All data and analysis code are available on GitHub (https://github.com/aschwalbc/MtbInf).

**Funding:** This work was supported by the European Research Council (https://erc.europa.eu/) [grant number 757699 to AS, KCH, and RMGJH]. The funders had no role in study design, data collection and analysis, decision to publish, or preparation of the manuscript.

**Competing interests:** I have read the journal's policy and the authors of this manuscript have the following competing interests: PJD is a paid statistical consultant on PLOS Medicine's statistical board. All other authors declare no competing interests.

**Abbreviations:** ARI, annual risk of infection; BCG, *Bacillus* Calmette Guérin; GATHER, Guidelines for Accurate and Transparent Health Estimates Reporting; GP, Gaussian process; HIV, human immunodeficiency virus; IGRA, interferon-gamma release assays; LTBI, latent TB infection; *Mtb*, *Mycobacterium tuberculosis*; TB, tuberculosis; TST, tuberculin skin test; UI, uncertainty interval; WHO, World Health Organization.

had the highest burden, with 39.1 million (95% UI: 18.0, 73.6), 12.0 million (95% UI: 5.8, 22.9), and 11.2 million (95% UI: 5.0, 25.5) people, respectively, recently infected with *Mtb*. Sensitivity analyses of varying self-clearance scenarios showed significant changes in global estimates of viable *Mtb* infection, particularly in total burden, with lower self-clearance rates. Overall uncertainty in the estimates was considerable, reflecting limitations in the underlying data informing key model parameters.

## Conclusions

Our findings offer global burden estimates of viable *Mtb* infection and reveal a sizable population recently infected with *Mtb* and at high risk of progression to disease. New diagnostic tools that can detect individuals with viable *Mtb*—particularly those who would benefit from TB preventive therapy—are urgently needed.

---

## Author summary

### Why was this study done?

- Estimates of *Mycobacterium tuberculosis* (*Mtb*) infection reflect immunoreactivity, rather than the presence of viable bacteria.

- As most people who develop tuberculosis (TB) do so within two years of infection, capturing the recency of infection improves the relevance of burden estimates.

- Estimating recent *Mtb* infection supports targeted TB prevention strategies.

### What did the researchers do and find?

- We developed a mathematical model that incorporated reversion-adjusted and age-specific annual risk of infection trends and accounted for self-clearance of *Mtb* infection.

- We estimated that around 134 million people—about 1.7% of the global population—carried viable *Mtb* from a recent infection in 2022.

- Nearly 87% of all recent infections were estimated to be in the World Health Organization regions of South-East Asia, the Western Pacific, and Africa.

- Sensitivity analyses reveal that our estimates are sensitive to varying assumptions about self-clearance rates.

### What do these findings mean?

- Our findings provide national, regional, and global estimates of *Mtb* infection burden, emphasising the distinction between viable infection and immunoreactivity.

- A diagnostic test that can detect viable *Mtb* infection is urgently needed to identify those who would benefit most from TB preventive therapy.

- Future estimates should address the main limitations of this study by using more contemporary, empirically derived annual risk of infection data through repeated immunoreactivity surveys, along with empirical measures of self-clearance.

## Introduction

Estimates of *Mycobacterium tuberculosis* (*Mtb*) infection burden are fundamental in shaping global health strategies for tuberculosis (TB) [1]. These estimates guide key interventions, such as the provision of TB preventive therapy (TPT) to individuals infected with *Mtb* (in the absence of disease), which is vital for reducing TB incidence [2]. Therefore, estimates must capture the number of individuals harbouring viable *Mtb* infection, i.e., an infection capable of causing disease [3]. Furthermore, these estimates should account for the recency of infection, as most individuals who progress to disease do so within two years of infection [4]. Focussing on recent viable *Mtb* infection sets a medically actionable target—serving as a programmatically relevant upper bound—for optimising the prevention cascade and the population to target with TPT [1,3]. Additionally, such estimates would offer deeper insights into the reservoir fuelling ongoing *Mtb* transmission in the coming years [1].

Previous studies have estimated that a substantial portion of the global population had 'latent' TB infection (LTBI), an asymptomatic state defined by the presence of immunoreactivity to tuberculous antigens [5–7]. Although these estimates essentially reflect individuals exposed to *Mtb* and still exhibiting immunoreactivity, they are often used interchangeably to indicate current infection [1,8]. As our understanding of the natural history of TB has evolved [9,10], long-standing assumptions that informed those estimates warrant reconsideration.

Firstly, estimates of LTBI used the annual risk of infection (ARI) as a metric for the force of infection experienced by a population. The ARI is derived from surveys of immunoreactivity prevalence, and assumes immunoreactivity persists over time [11]; however, immunoreactivity can wane and in several cases revert [12], leading to significant underestimation of the actual ARI [13,14]. Secondly, the ARI is often extrapolated from surveys in children [11]; however, *Mtb* transmission to children is less common than in adolescents and adults [15]. As a result, infection incidence (and the estimated ARI) may be underestimated in older age groups, who have higher contact rates with individuals with infectious TB [14,15]. Both reversion and contact patterns suggest that the true force of infection in adults is likely higher than previously assumed. If so, this would significantly increase the estimated global burden of infection, especially given the long-held assumption of lifelong *Mtb* infection. However, this assumption has been challenged by estimates suggesting that a large proportion (>90%) of individuals self-clear infection without treatment and are no longer at risk of TB in the absence of reinfection [8,9,16,17]. Therefore, accounting for self-clearance in our estimates would likely yield a lower global burden of infection than previously thought [9].

These progressive insights underscore the need for estimates that account for the dynamic nature of *Mtb* infection. By incorporating these factors, more accurate assessments of the current global burden of viable *Mtb* infection can be made, particularly to guide effective TB prevention strategies. Additionally, improved characterisation of viable *Mtb* infection—as part of the ongoing pathological process within the spectrum of TB disease—is needed to better identify and target individuals in whom typical bacteriological confirmation is not possible. In this study, we estimate the global burden of viable *Mtb* infection using a mathematical modelling approach that incorporates recent insights into TB natural history.

## Methods

### Annual risk of infection

To estimate the burden of viable *Mtb* infection, we constructed national ARI trajectories spanning from 1950 to 2022, based on the methods used by Houben and Dodd [6]. The trajectories were constructed using a Gaussian process (GP)

regression, a flexible, non-parametric framework combining different sources of estimates with the assumption of a normal approximation to the likelihood. These were fitted to 171 countries (comprising 99.6% of the world population) using two sources of ARI estimates. Direct ARI estimates were obtained from nationally representative immunoreactivity surveys identified in previous searches (**Table A** and **Text A in** S1 File). Most surveys used tuberculin skin test (TST) positivity prevalence and were conducted in children aged 6–9 years old. For estimates reported as a single value without pre-sentation of uncertainty, an additional step was taken to quantify measurement precision (**Text B in** S1 File). Moreover, indirect ARI estimates were derived from TB prevalence estimates using the revised Stýblo rule and adjusted to account for the influence of age and human immunodeficiency virus (HIV) on smear positivity (**Text C in** S1 File). TB prevalence estimates were calculated by converting the most recent World Health Organization (WHO) TB incidence estimates (from 2000 to 2022) by applying an average duration of disease (**Text D in** S1 File). For country-years in which both direct and indirect ARI estimates were available, these are compared in **Table B in** S1 File. Furthermore, accounting for the impact of immunoreactivity reversion and that the true ARI is roughly three times higher than calculated, all ARI estimates were adjusted based on this underestimation [13]. Additionally, measurement uncertainty was also increased by 50% to account for this adjustment (**Text E in** S1 File). The GP regression with a linear trend was applied to the reversion-adjusted data on ARI (on a log scale) and the measurement precision per country. For each country, we generated 1,000 simulated ARI trajectories spanning 1950–2022 (**Figs A–F in** S1 File).

## Mixing and age-specific risks of infection

To address the higher risk of infection in adults compared to children, we used age-specific estimates of TB incidence as a proxy for TB prevalence and then applied contact mixing matrices to derive the corresponding hazard ratios for each age group. We defined three distinct age groups to encapsulate varying ARIs: under 15 years, 15–45 years, and 45 years and older. Using the estimated TB incidence disaggregated by age and country [18], we calculated the relative TB incidence per capita, using the under-15-year-old age group as the reference, assuming that this age group contributes minimally to transmission (**Fig G in** S1 File). Similarly, by employing the synthetic country-specific contact mixing matrices developed by Prem and colleagues [19], we obtained the average number of contacts between the defined age groups (**Fig H in** S1 File). We then integrated these data to estimate the relative ARI for each age group, again using the under-15-year-old age group as the reference (**Fig I in** S1 File). Finally, ARI trajectories were subdivided into three age groups and adjusted based on relative ARI; for countries with missing data, we applied the regional relative ARI.

## Self-clearance rates

Using a Bayesian approach, self-clearance rates were calibrated by tracking an infected cohort over time, using a sim-plified version of the model structure described below. This structure comprised sequential infected compartments rep-resenting increasing time since infection, allowing for self-clearance while excluding infection or reinfection. Informed by the pathways described in the TB natural history model by Horton and colleagues [10], the proportions of the cohort that self-clear or recover (i.e., no longer harbour viable *Mtb* infection) were 80.9% (95% uncertainty interval [UI]: 65.1, 90.6%) at year 1, 91.9% (95% UI: 83.3, 96.3%) at year 2, and 97.2% (95% UI: 93.9, 98.7%) at year 10, which were used as cali-bration targets. Due to the lack of data on the proportion after 10 years, we assumed a proportion of 99.0% (95% UI: 97.8, 99.6) and calibrated the associated post-10-year self-clearance rate under two different scenarios: one where this was reached by year 20 post-infection (high self-clearance) and another for year 50 (low self-clearance) (**Fig J in** S1 File). The model by Horton and colleagues captures detailed progression and regression across multiple TB states, ranging from *Mtb* infection through asymptomatic and symptomatic infectious disease. As such, the complement of these calibration targets (i.e., the proportion that does not self-clear or recover), and thus our estimates, represent individuals who harbour viable *Mtb* bacilli, including those with TB disease [20]. Furthermore, in this study, the term self-clearance is used as an overarching concept encompassing both biological clearance without progression and natural recovery after progression

to disease. All rates were assigned uninformed uniform priors and posterior estimates were calculated using a Markov chain Monte Carlo algorithm in Stan via R statistical software [21,22]. We assumed that self-clearance rates were constant and were not influenced by age or year. Further details on calibration are available in **Text F in** S1 File.

## Model structure

We developed a deterministic model tracking *Mtb* infection and self-clearance of infection across 5-year age groups (**Fig K in** S1 File). Parameters and descriptions can be found in **Table C in** S1 File. Once individuals are infected, they can progress through four infection states reflecting various times since infection. In each infection state, individuals can self-clear infection and revert to not being infected. Individuals who have been distally infected (i.e., more than two years) are at risk of reinfection, adjusted by a protection factor [23]. The force of infection was determined by the reversion-adjusted and age-group-specific ARI trajectory for each country. The self-clearance rates were obtained from the calibrated scenarios. No correlations were assumed between parameters, except in a designated sensitivity analysis. The model was constructed using R version 4.3.2 for statistical computing and graphics [22]. Further details and model equations are available in the **Texts G and H in** S1 File.

## Model run and outputs

For each country, all 1,000 ARI trajectories from 1950 to 2022 were used to estimate the proportion of each age group harbouring viable *Mtb* at various times since infection. These proportions were combined with population estimates obtained from the United Nations World Population Prospects [24]. Estimates for absolute numbers and prevalence of viable *Mtb* infection were summarised and explored by age group, country, region, and globally. For the main results, we report the estimates based on the high self-clearance scenario as a conservative approach to estimating viable *Mtb* infection in 2022, with robustness compared against the low self-clearance scenario.

## Statistical analysis

All analyses were performed using R [22]. Model outputs are summarised and reported as medians with corresponding 95% UI, calculated as the 2.5%–97.5% percentile range.

## Sensitivity analyses

We conducted two sensitivity analyses to assess the impact of assumptions around self-clearance. First, we assumed that self-clearance parameters were correlated across countries: for each model run, a sampled set of self-clearance values was drawn and applied uniformly to all countries, reflecting the assumption that natural history processes may be consistent across settings. Second, we conducted a parameter-driven analysis in which self-clearance rates were systematically scaled by ±25%, ±50%, and ±75% relative to the median estimates from the calibrated high self-clearance scenario to explore how these rates influence global burden estimates.

## GATHER reporting

This study was reported in accordance with the Guidelines for Accurate and Transparent Health Estimates Reporting (GATHER) [25]. Fig 1 provides a comprehensive conceptual overview of the study, including methods and data sources. The GATHER checklist is also available in the S2 File.

## Results

### Annual risk of infection estimates

Fig 2 shows the fitted ARI trajectories with and without reversion and age-specific adjustments for India, Indonesia, and China—countries with the highest estimated TB incidence in 2022. Compared with the under-15-year-old population,

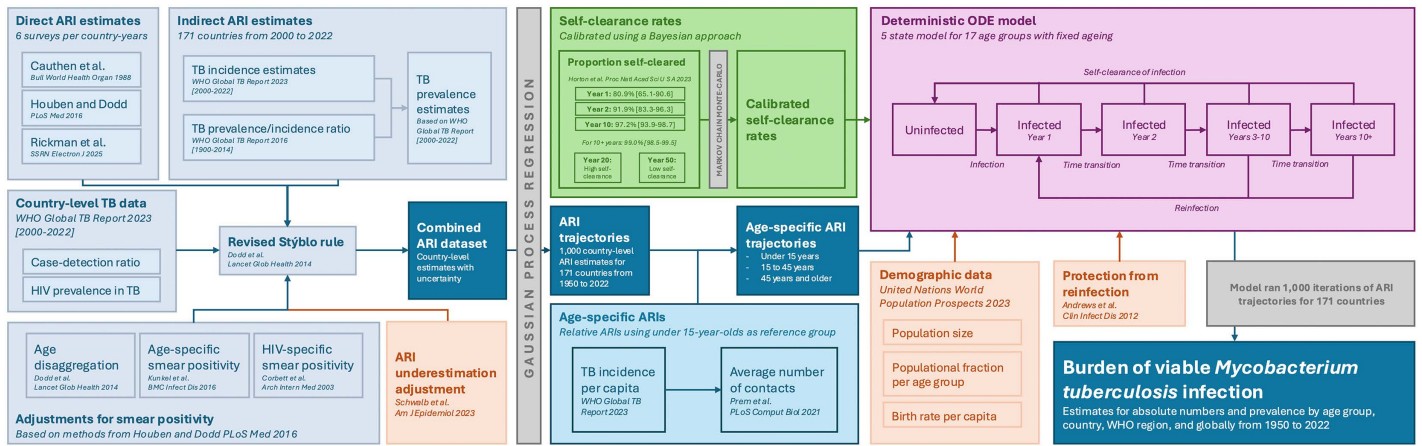

**Fig 1. Diagram of the conceptual overview of the study.** Conceptual overview of study as per Guidelines for Accurate and Transparent Health Estimates Reporting (GATHER) [25]. This diagram summarises the key data sources, analytical steps, and modelling components used to estimate the global burden of viable *Mycobacterium tuberculosis* infection. The left panels outline the derivation of ARI data: direct estimates from immunoreactivity surveys and indirect estimates derived from WHO TB incidence data via a revised Stýblo rule, with adjustments for smear positivity in children and people living with HIV. All ARI estimates were further adjusted for underestimation due to immunoreactivity reversion. The combined ARI dataset was smoothed using Gaussian process and stratified by age using TB incidence per capita and contact mixing matrices. These age-specific ARIs informed the force of infection input in the ODE model. Additional inputs included self-clearance rates (calibrated via MCMC to time point-specific targets) and demographic data on population size and age structure. Model outputs include national, regional, and global estimates of viable *Mtb* infection by age group. ARI, Annual risk of infection; HIV, Human immunodeficiency virus; MCMC, Markov chain Monte Carlo; *Mtb*, *Mycobacterium tuberculosis*; ODE, Ordinary differential equation; TB, Tuberculosis; WHO, World Health Organization.

the average relative ARI was 3.3 (95% UI: 2.9, 3.9) for the 15- to 45-year-old population and 2.9 (95% UI: 2.5, 3.3) for those aged 45 and older. Regional relative ARIs by age group are available in **Table D in** S1 File. For India, Indonesia, and China, the estimated ARI for the 15-to-45-year age group was 8.4% (95% UI: 4.1, 16.2), 13.6% (95% UI: 7.1, 25.3), and 2.0% (95% UI: 0.9, 4.3), respectively. ARI estimates for 2022 for the remaining countries on WHO's list of 30 high TB burden countries are shown in **Table E in** S1 File. When compared to the unadjusted ARI estimates for 2014 from Houben and Dodd [25], the ARI for the under-15-year age group was 3.1 times higher (interquartile range: 2.5, 4.0), reflecting the increase in the force of infection due to adjustments for reversion underestimation (**Table F in** S1 File).

## Global burden of infection estimates

We estimate that 133.7 million people (95% UI: 104.0, 171.1) had a recent *Mtb* infection in 2022, representing 1.7% (95% UI: 1.3, 2.2) of the global population. In total, 288.9 million people (95% UI: 242.2, 342.7)—or 3.7% (95% UI: 3.1, 4.3) globally—were harbouring a viable *Mtb* infection. Estimates of both total and recent infection were robust across scenarios, with minimal differences observed between the high and low long-term self-clearance assumptions (Table 1 versus **Table G in** S1 File for absolute numbers; Table 2 versus **Table H in** S1 File for proportions of the population).

## Regional and country-level infection estimates

Of all recent infections, approximately 87% were found in the three WHO regions of South-East Asia (49.0%; 95% UI: 37.2, 62.4), the Western Pacific (19.7%; 95% UI: 12.6, 30.5), and Africa (17.9%; 95% UI: 12.9, 24.1) (**Table I in** S1 File). At the country level, India, Indonesia, and China collectively represented around 50% of global recent infections, with 39.1 million (95% UI: 18.0, 73.6), 12.0 million (95% UI: 5.8, 22.9), and 11.2 million (95% UI: 5.0, 25.5) people recently infected, respectively (**Table J in** S1 File). The remaining countries in the top 10 were all from WHO's list of 30 high TB burden countries [26]. Fig 3 shows the regional and country-level variation in the prevalence of recent viable *Mtb* infections. Here,

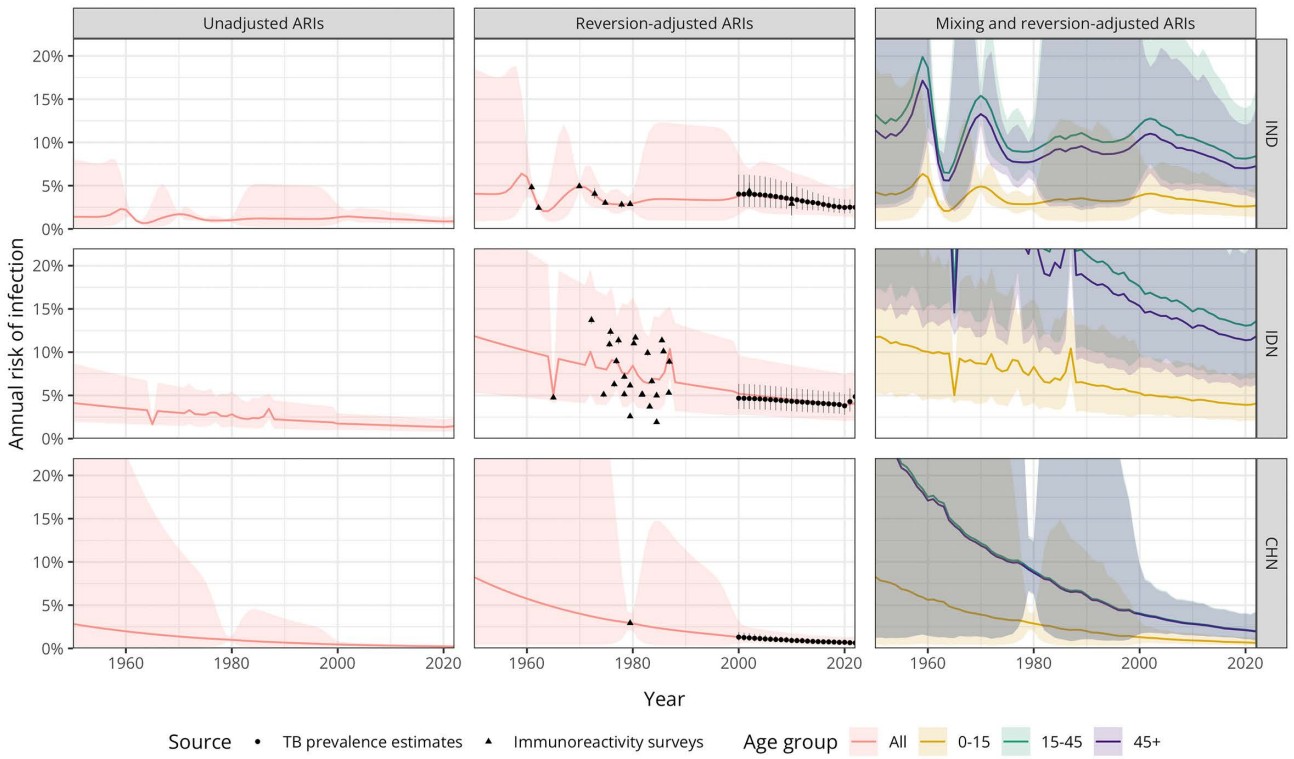

**Fig 2. Fitted annual risk of infection trajectories and adjustments.** Fitted annual risk of *Mycobacterium tuberculosis* infection trajectories for selected countries and corresponding adjustments. The first column shows ARI trajectories without adjustment, the second column shows trajectories adjusted for immunoreactivity reversion, and the third column shows the ARI trajectories subdivided by age group, using children under 15 years of age as the reference group for relative ARIs. Greater uncertainty is observed in earlier years, which narrows as ARI data becomes available from immunoreactivity surveys or TB prevalence estimates. Data points represent available ARI data, with black circles representing TB prevalence estimates and black triangles representing nationally representative immunoreactivity surveys; error bars reflect measurement precision to ±one standard deviation. Lines represent the mean ARI, and the shaded area shows ±one standard deviation from the Gaussian process regression with a linear trend. ARI, Annual risk of infection; CHN, China; IDN:, Indonesia; IND, India.

the countries with the highest prevalence of recent viable *Mtb* infections were the Philippines (7.7%; 95% UI: 3.2, 16.4), North Korea (5.4%; 95% UI: 2.2, 12.0), and Timor-Leste (5.3%; 95% UI: 2.2, 11.7) (**Table K in** S1 File). Estimates of viable *Mtb* infection for all 171 countries are available in the S1 Data.

### Age trends

The proportions of individuals with viable *Mtb* infection by age group and region are shown in Fig 4 and **Table L in** S1 File. Across all regions, higher infection estimates are concentrated in the adult population younger than 45 years old, in line with the increased ARI. Among those recently infected, 16.1 million (95% UI: 12.7, 20.3) were children under 15 years of age, representing 12.0% (95% UI: 11.4, 12.7) of all recent infections (Tables 1 and 2). There was a substantial regional disparity in recent infections among children, ranging from 7.9% (95% UI: 7.3, 8.6) in the European Region to 18.3% (95% UI: 17.6, 19.0) in the African Region, likely due to differences in population structure (Table 2).

### Sensitivity analyses

Under the scenario assuming correlated self-clearance rates across countries, the estimated burden of viable *Mtb* infection in 2022 was similar to the main analysis, though with wider UIs: 287.2 million people (95% UI: 196.1, 393.8) in total,

**Table 1. Number of individuals with viable *Mycobacterium tuberculosis* infection in 2022.**

| WHO region | Recent infections | Recent infections in children | All infections | All infections in children |
|---|---|---|---|---|
| | (M) [95% UI] | | | |
| AFR | 24.0 [19.0, 30.2] | 4.4 [3.4, 5.7] | 48.3 [41.7, 56.5] | 7.6 [6.3, 9.3] |
| AMR | 3.9 [2.8, 5.5] | 0.4 [0.3, 0.5] | 9.0 [7.1, 12.3] | 0.6 [0.5, 0.8] |
| EMR | 9.0 [5.3, 15.4] | 1.6 [0.9, 2.7] | 19.0 [13.4, 27.6] | 2.7 [1.8, 4.2] |
| EUR | 3.2 [2.4, 4.4] | 0.3 [0.2, 0.3] | 10.0 [8.1, 12.5] | 0.5 [0.4, 0.6] |
| SEA | 65.5 [41.6, 103.3] | 6.7 [4.2, 10.9] | 138.0 [100.0, 188.0] | 12.2 [8.5, 17.2] |
| WPR | 26.3 [16.7, 42.3] | 2.5 [1.6, 4.2] | 61.7 [44.8, 87.4] | 4.4 [3.2, 6.5] |
| **GLOBAL** | **133.7 [104.0, 171.1]** | **16.1 [12.7, 20.3]** | **288.9 [242.2, 342.7]** | **28.3 [23.7, 33.7]** |

Absolute number of individuals globally and by WHO region infected with viable *Mycobacterium tuberculosis* in 2022. Numbers are in millions (M), with brackets indicating 95% uncertainty intervals (UI). Recent infections are defined as those occurring within the past two years. Children are classified as individuals under 15 years of age. All estimates reflect the high self-clearance scenario. WHO, World Health Organization; AFR, African Region; AMR, Region of the Americas; EMR, Eastern Mediterranean Region; EUR:, European Region; SEA, South-East Asia Region; WPR, Western Pacific Region.

**Table 2. Proportion of population with viable *Mycobacterium tuberculosis* infection in 2022.**

| WHO Region | Recent infection prevalence | Proportion of recent infections in children | All infection prevalence | Proportion of all infections in children |
|---|---|---|---|---|
| | (%) [95% UI] | | | |
| AFR | 2.0 [1.6, 2.6] | 18.3 [17.6, 19.0] | 4.1 [3.5, 4.8] | 15.6 [14.9, 16.5] |
| AMR | 0.4 [0.3, 0.5] | 9.3 [8.8, 9.8] | 0.9 [0.7, 1.2] | 7.1 [5.6, 7.8] |
| EMR | 1.2 [0.7, 2.0] | 17.2 [16.7, 17.9] | 2.5 [1.7, 3.6] | 14.4 [13.0, 15.7] |
| EUR | 0.3 [0.3, 0.5] | 7.9 [7.3, 8.6] | 1.1 [0.9, 1.3] | 4.8 [3.8, 5.6] |
| SEA | 3.2 [2.0, 5.0] | 10.3 [10.1, 10.6] | 6.7 [4.8, 9.1] | 8.8 [8.1, 9.5] |
| WPR | 1.4 [0.9, 2.2] | 9.5 [8.1, 11.4] | 3.2 [2.3, 4.5] | 7.2 [5.6, 8.9] |
| **GLOBAL** | **1.7 [1.3, 2.2]** | **12.0 [11.4, 12.7]** | **3.7 [3.1, 4.3]** | **9.8 [9.1, 10.4]** |

Proportion of population globally and by WHO region infected with viable *Mycobacterium tuberculosis* in 2022. Values are given percentages (%), with brackets indicating 95% uncertainty intervals (UI). Recent infection is defined as occurring within the past two years. Children are classified as individuals under 15 years of age. All estimates reflect the high self-clearance scenario. WHO, World Health Organization; AFR, African Region; AMR, Region of the Americas; EMR, Eastern Mediterranean Region; EUR, European Region; SEA, South-East Asia Region; WPR, Western Pacific Region.

and 132.2 million people (95% UI: 79.5, 202.2) with recent infection (**Table M in** S1 File). Under varying self-clearance scenarios, global estimates of viable *Mtb* infection increased with lower self-clearance rates, particularly for total burden (Fig 5 and **Table N in** S1 File). The proportion of infections that were recent declined as the total burden rose accordingly. At higher self-clearance levels, total infection estimates remained relatively stable.

## Discussion

Our findings provide an estimate of the global burden of *Mtb* infection, with approximately 134 million people recently infected in 2022 and therefore at immediate risk of developing TB. In total, 289 million people—3.7% of the world's population—were harbouring a viable *Mtb* infection. These estimates show significant geographical variability, with infections concentrated in WHO regions of South-East Asia, the Western Pacific, and Africa. These findings underscore the need for enhanced diagnostic and management strategies to identify individuals with viable *Mtb* infection—particularly those who may benefit from TPT.

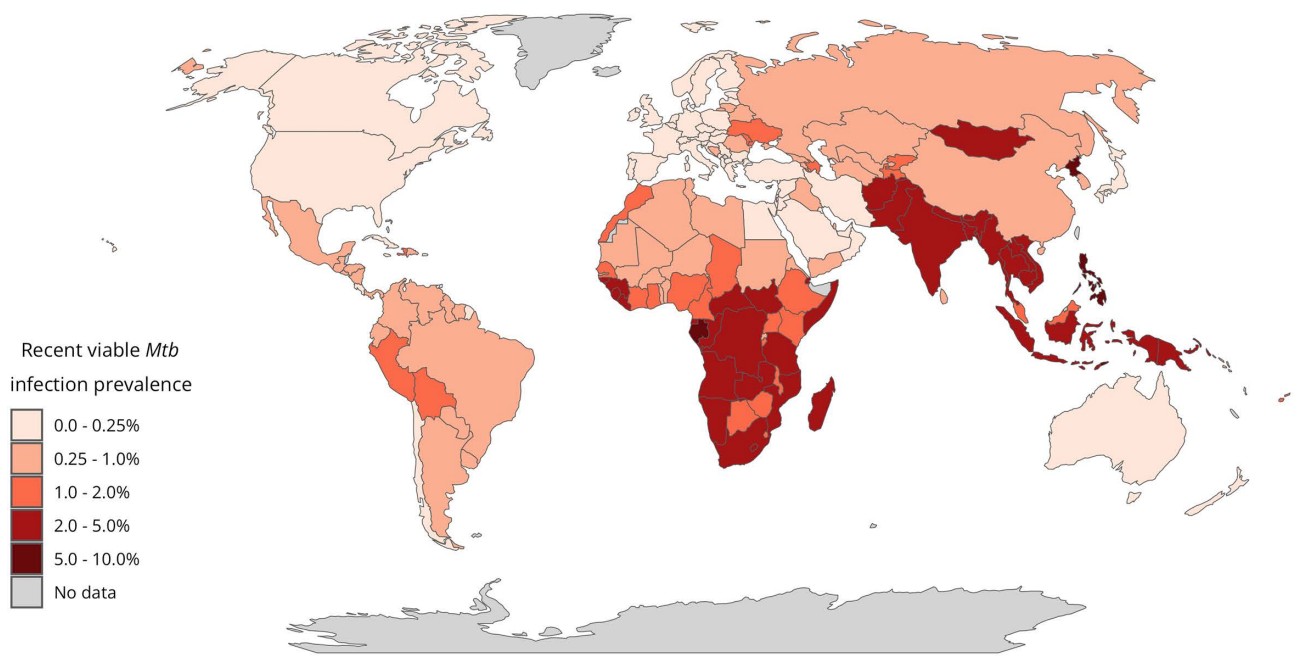

**Fig 3. Prevalence of recent viable *Mycobacterium tuberculosis* infection in 2022.** Median estimated population prevalence of recent viable *Mycobacterium tuberculosis* infection by country in 2022. Recent infection is defined as occurring within the past two years. Vector map data obtained from Natural Earth (https://www.naturalearthdata.com/).

Our estimates further highlight the important distinction between viable *Mtb* infection and positive immunoreactivity tests. Despite working on a similar construction of ARI trajectories, there is a stark numerical contrast between our viable *Mtb* infection estimates and the latest LTBI estimates [9]. The main contributor to this difference is the inclusion of self-clearance of infection in our model, thereby shifting the estimate definition from (historical) exposure to *Mtb* (as measured by immunoreactivity) to viable *Mtb* infection. This was suggested by Emery and colleagues, where the population with viable *Mtb* infection was markedly smaller (up to 20%) than assumed in India, China, and Japan [1,18]. Similarly, our estimates indicate a lower overall proportion of *Mtb* infection, even after adjusting for the increased ARI experienced (considering immunoreactivity reversion and age-specific risks). Self-clearance is now widely acknowledged, as reflected by a shift in WHO reporting, which now describes the quarter of the population figure as "having been infected" rather than "currently infected" [13,14]. Self-clearance, therefore, outweighed the increased force of infection generated by a revised understanding of the true ARI underneath empirical measurements [6]. However, the shift in ARI has led to a higher estimate of the recently infected population than previous estimates (1.7 versus 0.8%), suggesting a more rapid turnover of the population at high risk of developing TB [12].

Although reversion and self-clearance are likely related phenomena, they represent conceptually distinct processes in the context of this analysis. Reversion, defined as the loss of immunoreactivity [13,14], can lead to the underestimation of ARI if not accounted for, as individuals with prior infection may test negative [8,9,16,17]. In contrast, self-clearance refers to the biological elimination of viable *Mtb* bacilli from the body [8]. While reversion may occur in individuals who have self-cleared, not all reversion indicates true clearance, nor does all self-clearance necessarily result in reversion [26].

While this study attempts to quantify the extent of viable *Mtb* infection, our ability to detect it remains hampered by the limitations of currently available tests. These include the TST and interferon-gamma release assays (IGRA), which detect immunoreactivity to tuberculous antigens but do not directly detect the presence of the organism itself [27]. Another

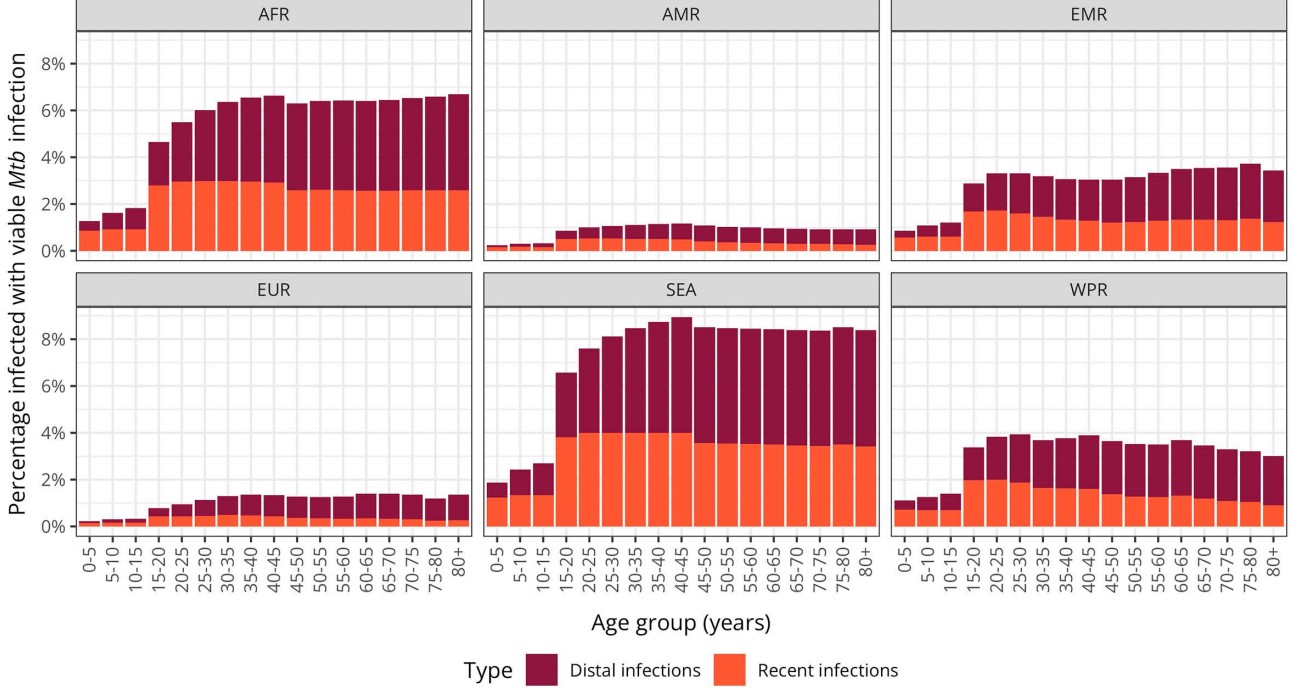

**Fig 4. Prevalence of viable *Mycobacterium tuberculosis* infection by age and WHO region in 2022.** Median estimated proportion of the population per age group and by WHO region infected with viable *Mycobacterium tuberculosis* in 2022. Values are presented as percentages (%). Recent infections are defined as those occurring within the past two years, while distal infections are those acquired more than two years ago. The coarse age group distribution in the estimates reflects the annual risk of infection disaggregated into the following groups: under 15 years old, 15–45 years old, and 45 years and older. Estimates are based on a scenario assuming high long-term self-clearance rates. WHO, World Health Organization; AFR, African Region; AMR, Region of the Americas; EMR, Eastern Mediterranean Region; EUR, European Region; SEA, South-East Asia Region; WPR, Western Pacific Region.

limitation can be observed in individuals with non-specific chest radiography abnormalities who are treated for TB despite lacking bacteriological confirmation and clear evidence of *Mtb* infection. Additionally, current immunoreactivity tests are inaccurate surrogates of viability as they can remain detectable (positive) after TPT or treatment [28]. This poses a significant challenge for national TB programmes, as TPT is often offered to those with a single positive immunoreactivity test [29], whereas the number of those that would truly benefit is likely smaller, especially in absence of an indicative patient history, such as recent contact with a TB patient. To improve and expand the use of TPT—and to aid in the diagnosis of bacteriologically negative TB disease—there is a clear need for improved biomarkers of *Mtb* infection, some of which have been explored recently [30,31].

Our modelling study is not without limitations. A major limitation concerns the estimation of the ARI. Empirical ARI estimates remain scarce, so contemporary ARIs had to be inferred indirectly from WHO TB incidence estimates, which provided the necessary data points for the GP regression. In country-years where both direct and indirect ARI estimates were available, more than half had non-overlapping uncertainty intervals; however, the GP regression, as a non-parametric framework, was able to accommodate this discrepancy. Ideally, population-level trends in *Mtb* transmission should be evaluated through repeated immunoreactivity surveys to obtain direct ARI estimates. While still requiring adjustments to account for reversion, recent surveys using IGRA or new TB antigen-based skin tests would improve ARI estimates, especially as they reduce false-positive results among children who received *Bacillus* Calmette Guérin (BCG) vaccination [27].

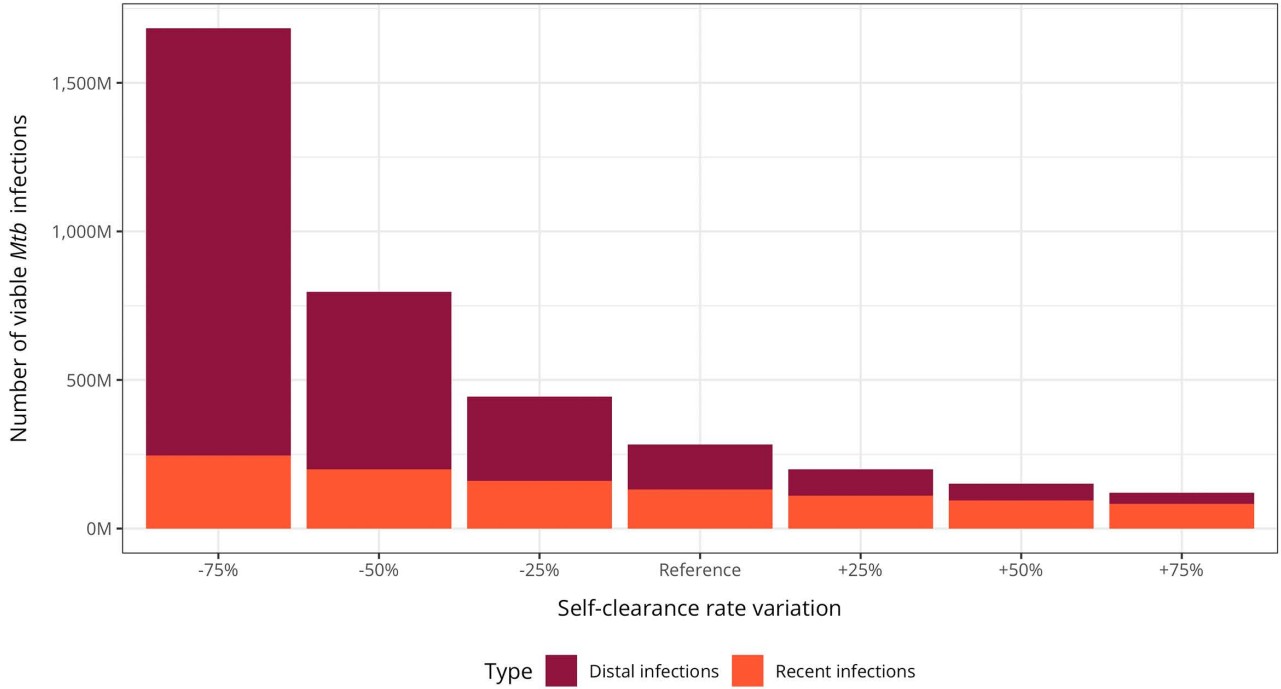

**Fig 5. Global estimates of viable *Mycobacterium tuberculosis* infection under varying self-clearance scenarios.** Absolute number of individuals infected with viable *Mycobacterium tuberculosis* in 2022. Values are shown in millions (M) with error bars indicating 95% uncertainty intervals (UI). Recent infections are defined as those occurring within the past two years, while distal infections are those acquired more than two years ago. Estimates reflect systematic variation of self-clearance rates relative to the median estimates from the high self-clearance scenario.

Another major limitation concerns the evidence base for self-clearance. Although a few studies summarise supportive evidence [1,4,8,9], empirical data remain limited. As our results show, assumptions about self-clearance are a major source of uncertainty in our estimates. Our sensitivity analysis on systematically varied self-clearance rates showed that global burden estimates were particularly sensitive to lower self-clearance rates, highlighting the need for better empirical data to inform this parameter. Additionally, these self-clearance rates were applied uniformly, but they may plausibly vary by age and HIV status; we currently lack the data to inform this heterogeneity. Future work should explore and incorporate such variation where possible.

Furthermore, several broader modelling simplifications were required to generate global estimates. We treated all individuals as experiencing one of three age-specific ARIs, disregarding population and individual factors that may affect the force of infection. ARI estimates could have been further refined by breaking them down into distinct 5-year age groups; however, given that self-clearance had the strongest effect on model outputs, we do not expect that adding further granularity to the ARI estimates would have changed the qualitative findings. Additionally, the reversion underestimation factor was simplified to a single value and applied uniformly to all ARI estimates, despite known variations across ages and populations [31]. We accounted for reduced measurement precision to reflect the additional uncertainty introduced by this assumption. Furthermore, although our model does not explicitly include a compartment for individuals who have self-cleared infection, we incorporated protection against reinfection by reducing the force of infection for individuals with current infection; however, this may underestimate the potential contribution of immunological memory in those who have cleared infection. Finally, because the calibration targets for the proportion self-cleared or recovered were derived from a model that accounts for progression and regression across multiple TB disease states and does not include

treatment-driven recovery, our estimates of viable *Mtb* infection include individuals with TB disease. As a result, the burden, particularly the fraction that might be considered medically actionable, may represent an upper bound. We recognise that addressing this limitation fully would require a more complex model, incorporating explicit representation of multiple TB disease states and consideration of treatment-driven recovery. We see this as an important and natural direction for future work. Nevertheless, while these necessary simplifications may have introduced some biases in our estimates, they were appropriate to ensure the feasibility of assessing *Mtb* infection burden at national, regional, and global levels.

Our findings provide valuable global estimates of the burden of viable *Mtb* infection, emphasising the crucial distinction from positive immunoreactivity tests. By more accurately accounting for the true force of infection and the immune system's ability to clear infections, we identified a sizable population recently infected and at high risk of progressing to disease. This approach reinforces the importance of better characterising viable *Mtb* infection within the spectrum of TB disease [20], as opposed to the concept of 'latency', which imposes a rigid separation from disease and relies on the notion of dormant infection to explain future risk. There is an urgent need for enhanced diagnostic tools capable of detecting viable *Mtb* infection, allowing us to identify individuals who may benefit from TB preventive therapy, as well as those with bacteriologically unconfirmed disease who might otherwise go undiagnosed.

## Supporting information

**S1 File. Supplementary material.** Additional methodological details, supplementary tables, and figures.
(PDF)

**S2 File. GATHER checklist.** GATHER checklist is made available under the Creative Commons Attribution 4.0 (CC BY 4.0) license. The original checklist can be accessed at: https://doi.org/10.1371/journal.pmed.1002056.
(PDF)

**S3 File. PMED Inclusivity in Global Research.** Checklist addressing ethical, cultural, and scientific considerations for inclusivity in global research.
(PDF)

**S1 Data. Country-level viable *Mycobacterium tuberculosis* infection estimates.** Country-level estimates of viable *Mycobacterium tuberculosis* infection, including recent and all infections, reported as absolute numbers and population prevalence, overall and among children.
(XLSX)

## Acknowledgments

The authors would like to acknowledge Prof. Marcel Behr and Prof. Dave Moore for their valuable insights on the methodology employed and estimates generated. Additionally, the authors extend their gratitude to Dr. Tomos Prŷs-Jones for his assistance in developing the pipeline to run the analyses on the high-performance computing platform. This research was supported in part through computational resources provided by the London School of Hygiene and Tropical Medicine.

## Author contributions

**Conceptualisation:** Alvaro Schwalb, Peter J. Dodd, Rein M. G. J. Houben.

**Data curation:** Alvaro Schwalb.

**Formal analysis:** Alvaro Schwalb, Peter J. Dodd.

**Supervision:** Peter J. Dodd, Katherine C. Horton, Rein M. G. J. Houben.

**Writing – original draft:** Alvaro Schwalb.

**Writing – review & editing:** Peter J. Dodd, Hannah M. Rickman, César A. Ugarte-Gil, Katherine C. Horton, Rein M. G. J. Houben.

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
