## [Editor Report · Decision Letter 0]

29 Nov 2024

Dear Dr Schwalb,

Thank you for submitting your manuscript entitled "Estimating the global burden of viable Mycobacterium tuberculosis infection" for consideration by PLOS Medicine.

Your manuscript has now been evaluated by the PLOS Medicine editorial staff as well as by an academic editor with relevant expertise and I am writing to let you know that we would like to send your submission out for external peer review.

Please re-submit your manuscript within two working days, i.e. by Dec 03 2024 11:59PM. However, please do not hesitate to let me know if you need more time (ssunny@plos.org).

Kind regards,

Syba

Syba Sunny, MBBS, MRes, FRCPath

Associate Editor

PLOS Medicine

---

## [Decision Letter · Decision Letter 1]

29 Jan 2025

Dear Dr Schwalb,

Many thanks for submitting your manuscript "Estimating the global burden of viable Mycobacterium tuberculosis infection" (PMEDICINE-D-24-03899R1) to PLOS Medicine. The paper has been reviewed by subject experts; their comments are included below and can also be accessed here: [LINK]

As you will see, the reviewers find the manuscript topical and timely, however both referees find that the manuscript would be strengthened by further analyses to better account for the uncertainty in key parameters, using a range of estimates for viable Mtb burden, self-clearance and reversion to immunoreactivity. Reviewer 1 questions why data from only one African country was used, and both referees raise concerns about the reliance on indirect annual risk of infection estimates, and would like you to clarify whether those match estimates derived from immunoreactivity for countries that have such data.

Please note that the reviewers also request specific clarifications to the Methods, further discussion of the study limitations and additional explanation of why you are unable to generate age (and HIV-status)-specific self-clearance estimates. their model? Please also discuss further the relationship between reversion and self-clearance. Please discuss whether your estimates are consistent with rates of reactivation in persons with immunosuppression due to therapy or living with AIDS, and if not why.

After discussing the paper with the editorial team and an academic editor with relevant expertise, I'm pleased to invite you to revise the paper in response to the reviewers' comments. We plan to send the revised paper to some or all of the original reviewers, and we cannot provide any guarantees at this stage regarding publication. Please note that we request that you indicate to the editors whether you considered the Global Code (https://www.globalcodeofconduct.org/) when developing, conducting, and authoring this research. PLOS has a 'Inclusivity in Global Research' policy which aims to promote collaboration and inclusivity in global health research. You are required to complete PLOS’ questionnaire on inclusivity in global research and submit it with your revised paper. The policy and questionnaire can be found at https://journals.plos.org/plosone/s/best-practices-in-research-reporting.

We ask that you submit your revision by Feb 19 2025 11:59PM. However, if this deadline is not feasible, please contact me by email, and we can discuss a suitable alternative.

Don't hesitate to contact me directly with any questions (afarrell@plos.org).

Best regards,

Alison

Alison Farrell, Ph.D.

Senior Editor

PLOS Medicine

afarrell@plos.org

Comments from the reviewers:

Reviewer #1: Thank you to the authors and editors for the opportunity to review this interesting manuscript. Schwalb et al conduct a modelling study to estimate the burden (and prevalence) of 'viable' Mtb infection. This is an important area for research, topical and timely in view of renewed interest in the natural history of TB. The manuscript is well written and the analyses are mostly clearly presented. The corresponding code are also presented in a GitHub repository, ensuring transparency. Even as a non-mathematical modeler, I was able to follow most of what was done, which is appreciated.

Overall, I think it is a very interesting idea to estimate the burden of viable Mtb infection. However, as the authors acknowledge, this is something that we cannot currently measure empirically (certainly not at scale). Thus, the authors have adopted a mathematical modelling approach. I am concerned that this relies heavily on estimates derived from previous modelling studies - which are themselves often conducted on suboptimal datasets. Thus, there is a huge amount of uncertainty in the key parameters used in this analysis which is not necessarily accounted for in the results. As a result, the results presented could actually be quite inaccurate and I am nervous about the idea of 'confidently' estimating the burden of viability, given that the uncomfortable truth is that we really don't know.

Thus, I would find the analysis much more compelling if this uncertainty was better accounted for. Once this is done, it might be the case that the range of estimates of the burden/prevalence of viable Mtb infection is very wide. If that is the case, then so be it… I think this would be preferable to a more precise, but potentially quite inaccurate estimate.

My primary concerns are itemised below.

1. Mtb clearance over time. The authors derive Mtb clearance estimates from Horton et al, PNAS 2023 (reference 10). Having reviewed the Horton et al paper, the estimate corresponds to Figure 2. From this, they consider that self-clearance occurs at years 1 (95% CI: 80.1-81.7%), 2 (95% CI: 91.4-92.5%), and 10 (95% CI: 96.9-97.5%). There is a little more uncertainty after year 10.

I am concerned about the precision of these estimates. E.g. how can we be confident that 80.1 - 81.7% of people self-clear by year 1 when this is impossible to measure empirically? For example, could it actually be that Mtb viability is often maintained after year 1 but the risk of disease progression declines? The Horton et al study is based on modelling of historic data. I find this level of precision far too narrow to be plausible. As Schwalb et al acknowledge, self-clearance is a primary determinant of their main results. Thus, I think it would be much more realistic to explore a wide range of self-clearance parameters (e.g. 50-90% at year 1… with similar uncertainty at subsequent years).

A more detailed summary of how these self-clearance values were estimated and the underlying assumptions would also be of value in the current manuscript, along with acknowledgment of ongoing uncertainty here as a major (perhaps even critical?) limitation.

2. Reversion of immunoreactivity. Following their previous work (Schwalb et al Am J Epidemiol. 2023; ref 13), the authors adjust their annual risk of infection estimates for reversion. They estimate that the ARI should be inflated x2-5 times to account for this. However, the exact multiplication factor is not presented in the manuscript. From the GitHub code, this appears to be 2.9. I again find this to be a very strong assumption, based on limited data. From reviewing the Am J Epidemiology paper, it seems that underestimation of the ARI was very heterogeneous by age and annual reversion probability (Figure 2 of that paper). Thus, a multiplication factor of 2.9 across the board seems difficult to justify. As for my point 1 above, why not consider a range of estimates (e.g. 2-5x) here to account for the huge amount of uncertainty? I note that in the discussion, the authors allude to age-specific adjustments being unfeasible. This seems like a shame.

3. In the introduction, the authors refer to viable Mtb infection as a 'medically actionable target'. But, as discussed later in the discussion, it isn't really medically actionable if we can't measure it? This speaks to a broader point about the entire analysis - what is the true value of such an analysis and estimation of burden when it is currently not possible to measure? I'd like to see a little more discussion around this, in addition to my points above on unaddressed uncertainty.

4. I am surprised that only 1 African country (DR Congo) appears in Table S7 (countries with highest prevalence of viable Mtb infection). Can the authors explain this? For example, South Africa has higher annual TB incidence rates than many of the countries listed in the table. Could there be a systematic issue with the analysis approach or underlying assumptions that has led to viable Mtb prevalence in African countries being underestimated?

5. As a more minor point, in the "Regional and country level infection estimates" section of the results the supplementary tables S5, S6 and S7 referenced in the text appear to be different to that in the supplementary material.

Reviewer #2: This study generated estimates of the global prevalence of viable Mtb infection and recent viable Mtb infection. The conclusions by the authors that infection prevalence is likely lower than previously estimated, but recent infection prevalence is higher, are important, and are a natural extension of our updated understanding of Mtb infection dynamics and the interpretation of immunoreactivity tests. Therefore I believe this study represents an important contribution to the TB epidemiology literature, and has practical implications for interventions and policy. I have a few comments and questions on the methodology, and suggestions for portions of the text that could be more clearly described.

Major Comments:

1. Many assumptions (about the relationship between incidence and prevalence, and then the relationship between prevalence and ARI, and also incidence itself is for many countries a model-based estimate) go into the indirect ARI estimates. Since few countries have recent immunoreactivity surveys, it appears that the indirect ARI estimates contribute heavily to the estimates of ARI in recent years in the GP regression. Were the authors able to do any validation of the indirect ARI approach? For example, do direct and indirect ARI estimates match, in country-years that have immunoreactivity surveys?

2. On a related note, I was surprised to see the narrowness of the uncertainty intervals presented in tables 1-2, given the large uncertainty intervals in the ARI estimates presented in table S4. I wonder if this may come down to much of the uncertainty getting "averaged out", as a result of how correlations in uncertain parameters (like ARI) are handled across age groups, countries, and years. For example, +/- 50% uncertainty was added to the ARI uncertainty intervals when the 3x adjustment was made. In the 1000 iterations, were higher ARI estimates in 1 country/year/age group paired with higher ARI estimates in other countries/years/age groups, or was each of the 1000 samples from a country-age-year ARI distribution treated as independent from the other country-age-year ARI distributions? It is reasonable to expect that if the 3x adjustment is an overestimate (or underestimate) in 1 setting, age group, or year, it could be an overestimate in other settings, age groups, and years, but also reasonable to expect that some countries would have a different adjustment factor than others (perhaps due to the characteristics of their survey, etc.). Could the authors please clarify how correlations between parameters were handled, and consider exploring sensitivity analysis that does include correlations in some parameters (or components of parameter estimation), if correlations are not currently included.

3. Figure 1 is generally very helpful, but some additional explanation (i.e., as a caption) would be appreciated and would make the methods much easier to understand. For example, what do the colors and light vs. dark shading represent? The left-hand side is the most unclear: do arrows between boxes indicate that one box led to another box, or that those two boxes were combined to produce a third box, or both? (for example, did the ARI estimates, country-level TB data, and adjustments for smear positivity all contribute to the revised Styblo rule, or were they combined with the revised Styblo rule to produce the combined ARI dataset? Was the Styblo rule actually applied to the Direct ARI estimates?)

4. Methods, self clearance rates and model structure sections: Why are progression and death not modeled as competing risks in either model? Presumably ignoring these competing risks would lead to an overestimation of the prevalence of infection.

5. Methods: 80-82% clearance after 1 year of infection is both very high and very certain. Could the authors please include sensitivity analysis that explores wider variation in self-clearance, or lower self-clearance targets? I don't doubt that self-clearance is higher than most models assume, but have trouble believing that we can describe self-clearance over time with this much precision.

6. Is there any reason why those who were infected and then self-cleared do not experience protection from reinfection in the model? Presumably some maintain immunoreactivity which could imply immunological memory? If it is uncertain whether this protection factor would apply, perhaps this could be mentioned as a limitation.

7. Reversion and self-clearance are almost certainty related phenomena, but they are decoupled in this analysis. Annual reversion of 15-30% from Schwalb et al. 2023 and annual clearance of 80% at year 1 from Horton et al. 2023 do not seem necessarily inconsistent, but a brief discussion of the relationship between reversion and self-clearance would be helpful for contextualizing the analysis.

Minor Comments:

8. Methods, self clearance rates: it is not clear what the "similar model structure as described above" is referring to. Is it the model structure in the following section?

9. Methods, self clearance rates: can the authors please clarify whether in this step they are calibrating a model to fit estimates of self-clearance that were generated from another model? If this is the case (a) can you please explain why you didn't just use that model's self-clearance rates and (b) where the year 1, 2, and 10 clearance targets are coming from in that paper (Horton et al. 2023)? In the results of that paper, "Over a 10-y period following Mtb infection, 92.0% (95% UI 91.4 to 92.5) of simulated individuals cleared infection without progressing to TB, with 90.5% (95% UI 89.9 to 91.1) clearing infection within the first 2 y" doesn't match the year 2 and 10 targets reported here.

10. Figure 1: how is the case-detection rate being used? I didn't see this described in the text or appendix but may have missed it.

11. Methods: Was the "revised Styblo rule" re-estimated for the present study (i.e., to include more recent publications and adjust for underestimation of ARI), or was the revised rule from Dodd et al. 2014 used as is? Could the appendix include a quantitative description of the revised rule?

12. Methods: What is the purpose of applying smear-positivity in the construction of the revised Styblo rule? Are the authors assuming that only smear-positive individuals are infectious?

13. Results, Global burden of infection estimates: Were the estimates of recently infected exactly the same across the 2 long-term clearance scenarios? I would expect slight differences due to the protection from reinfection factor (more clearance -> less protection -> more new infections), but maybe they round to the same values? Perhaps this could be clarified in the Table 1 and 2 captions.

14. I realize we probably lack the data to inform age- or HIV-status-specific self-clearance rates, but it seems biologically plausible that such variation could exist. Could some discussion of this be added?

---

* Please upload any figures associated with your paper as individual TIF or EPS files with 300dpi resolution at resubmission; please read our figure guidelines for more information on our requirements: http://journals.plos.org/plosmedicine/s/figures. While revising your submission, please upload your figure files to the PACE digital diagnostic tool, https://pacev2.apexcovantage.com/. PACE helps ensure that figures meet PLOS requirements. To use PACE, you must first register as a user. Then, login and navigate to the UPLOAD tab, where you will find detailed instructions on how to use the tool. If you encounter any issues or have any questions when using PACE, please email us at PLOSMedicine@plos.org.

* [EDITOR: CHECK FINANCIAL DISCLOSURES, COI, DAS, AND ETHICS STATEMENTS AND INCLUDE ANY NECESSARY REQUESTS]

* Please ensure that the study is reported according to the [XXXX] guideline and include the completed [XXXX] checklist as Supporting Information. When completing the checklist, please use section and paragraph numbers, rather than page numbers. Please add the following statement, or similar, to the Methods: "This study is reported as per [XXXX] guideline (S1 Checklist)."

FIGURES AND TABLES

SUPPLEMENTARY MATERIAL

REFERENCES

[STUDY TYPE-SPECIFIC REQUESTS - DELETE SECTIONS AS NECESSARY]

RCTs [REFER TO RCT CHECKLIST AND MEETING NOTES FOR DETAILS TO ADD]

* PLOS Medicine requires that all trials be prospectively registered in one of registries recognized by WHO. Please ensure that study registration details are included in the Methods section.

* Please structure the Methods section using the following sub-headings: Study design and participants, Randomization and masking, Procedures, Outcomes, Statistical analysis.

* The following outcomes measures [ADD DETAILS AS NEEDED OR DELETE BULLET POINT] appear to differ between the submitted manuscript and the protocol [and/or trial registry]. Please clarify and explain all discrepancies between the paper and protocol. If the outcomes were not prespecified in the protocol, please define them in the Methods (Outcomes section) as post hoc and explain why they were added. Post-hoc comparisons should be presented as hypothesis generating rather than conclusive.

* Please ensure that all prespecified outcomes (primary, secondary, and exploratory) are listed in the Methods/Outcomes section and indicate whether there are outcomes that are not presented in the current report.

* Please specify the dates (Month Day, Year) during which study enrollment and follow up occurred.

* Please include absolute numbers wherever you report percentages; eg, n/N (%)

* Please present the safety data for the study including numbers of specific events and whether or not adverse events are thought to be related to treatment. AEs should be reported in the abstract, per CONSORT and CONSORT-Harms.

* Please complete the CONSORT checklist (https://www.equator-network.org/reporting-guidelines/consort/) and ensure that all components of CONSORT are present in the manuscript, including how randomization was performed, allocation concealment, blinding of intervention, definition of lost to follow-up, power statement. When completing the checklist, please use section and paragraph numbers, rather than page numbers.

* Please report your abstract according to CONSORT for abstracts, following the PLOS Medicine abstract structure (Background, Methods and Findings, Conclusions) https://www.equator-network.org/reporting-guidelines/consort-abstracts/

* If your trial had to undergo important modifications in response to extenuating circumstances, please complete the CONSERVE-CONSORT checklist and provide in your Supporting Information; (https://www.equator-network.org/reporting-guidelines/guidelines-for-reporting-trial-protocols-and-completed-trials-modified-due-to-the-covid-19-pandemic-and-other-extenuating-circumstances-the-conserve-2021-statement/). When completing the checklist, please use section and paragraph numbers, rather than page numbers.

* In keeping with our commitment to Open Science, please include the study protocol document and analysis plan (including any amendments) as Supporting Information to be published with the manuscript if accepted.

* Please note that PLOS Medicine requires prospective, public registration of a data sharing plan (as part of mandatory clinical trials registration) for all clinical trials that began enrollment on or after January 1, 2019, in accordance with ICMJE requirements.

OBSERVATIONAL STUDIES

* Abstract: Please include the study design, population and setting, number of participants, years during which the study took place (enrollment and follow up), length of follow up, and main outcome measures.

* Please ensure that the study is reported according to the STROBE (or appropriate STOBE extension) guideline (available from: https://www.equator-network.org/reporting-guidelines/strobe) and include the completed STROBE (or STROBE extension) checklist as Supporting Information. Please add the following statement, or similar, to the Methods: "This study is reported as per the Strengthening the Reporting of Observational Studies in Epidemiology (STROBE) guideline (S1 Checklist)." When completing the checklist, please use section and paragraph numbers, rather than page numbers.

* [FOR POPULATION HEALTH/REGISTRY STUDIES] Please ensure that the study is reported according to the RECORD guideline (available from https://www.record-statement.org) and include the completed checklist as Supporting Information. Please add the following statement, or similar, to the Methods: "This study is reported as per the Reporting of Studies Conducted using Observational Routinely-Collected Data (RECORD) guideline (S1 Checklist)." When completing the checklist, please use section and paragraph numbers, rather than page numbers.

* [FOR POPULATION HEALTH ESTIMATES] Please ensure that the study is reported according to the GATHER statement (available from https://www.equator-network.org/reporting-guidelines/gather-statement) and include the completed checklist as Supporting Information. Please add the following statement, or similar, to the Methods: "This study is reported as per the Guidelines for Accurate and Transparent Health Estimates Reporting (GATHER) statement (S1 Checklist)." When completing the checklist, please use section and paragraph numbers, rather than page numbers.

* [FOR MEDIATION ANALYSES] We recommend that the study is reported according to the AGReMA statement (https://agrema-statement.org/#:~:text=AGReMA%20is%20an%20evidence%2D%20and,randomised%20trials%20and%20observational%20studies) and include the completed checklist as Supporting Information. Please add the following statement, or similar, to the Methods: "This study is reported as per the Guideline for Reporting Mediation Analyses (AGReMA) statement (S1 Checklist)." When completing the checklist, please use section and paragraph numbers, rather than page numbers.

* For all observational studies, in the manuscript text, please indicate: (1) the specific hypotheses you intended to test, (2) the analytical methods by which you planned to test them, (3) the analyses you actually performed, and (4) when reported analyses differ from those that were planned, transparent explanations for differences that affect the reliability of the study's results. If a reported analysis was performed based on an interesting but unanticipated pattern in the data, please be clear that the analysis was data driven.

* Please state in the Methods section whether the study had a prospective protocol or analysis plan. If a prospective analysis plan (from your funding proposal, IRB or other ethics committee submission, study protocol, or other planning document written before analyzing the data) was used in designing the study, please include the relevant document(s) with your revised manuscript as a Supporting Information file to be published alongside your study and cite it in the Methods section. A legend for this file should be included at the end of your manuscript. If no such document exists, please make sure that the Methods section transparently describes when analyses were planned, and when/why any data-driven changes to analyses took place. Changes in the analysis, including those made in response to peer review comments, should be identified as such in the Methods section of the paper, with rationale.

MODELLING STUDIES

The following list is derived from Geoffrey P Garnett, Simon Cousens, Timothy B Hallett, Richard Steketee, Neff Walker. Mathematical models in the evaluation of health programmes. (2011) Lancet DOI:10.1016/S0140-6736(10)61505-X:

* If pertinent, please provide a diagram that shows the model structure, including how the natural history of the disease is represented, the process and determinants of disease acquisition, and how the putative intervention could affect the system.

* Please provide a complete list of model parameters, including clear and precise descriptions of the meaning of each parameter, together with the values or ranges for each, with justification or the primary source cited and important caveats about the use of these values noted.

* Please provide a clear statement about how the model was fitted to the data, including goodness-of-fit measure, the numerical algorithm used, which parameter varied, constraints imposed on parameter values, and starting conditions.

* For uncertainty analyses, please state the sources of uncertainties quantified and not quantified [can include parameter, data, and model structure].

* Please provide sensitivity analyses to identify which parameter values are most important in the model. Uncertainty estimates seek to derive a range of credible results on the basis of an exploration of the range of reasonable parameter values. The choice of method should be presented and justified.

* Please discuss the scientific rationale for the choice of model structure and identify points where this choice could influence conclusions drawn. Please also describe the strength of the scientific basis underlying the key model assumptions.

* For studies that develop a prediction model or evaluate its performance, please ensure that the study is reported according to the TRIPOD statement (https://www.equator-network.org/reporting-guidelines/tripod-statement) and include the completed checklist as Supporting Information. Please add the following statement, or similar, to the Methods: "This study is reported as per the Transparent Reporting of a Multivariable Prediction Model for Individual Prognosis Or Diagnosis (TRIPOD) statement (S1 Checklist)." For studies using machine learning, please use the TRIPOD-AI checklist. When completing the checklist, please use section and paragraph numbers, rather than page numbers.

DIAGNOSTIC STUDIES

* Please ensure that the study is reported according to the STARD guideline (https://www.equator-network.org/reporting-guidelines/stard/) and include the completed STARD checklist as Supporting Information. Please add the following statement, or similar, to the Methods: "This study is reported as per the Standards for Reporting of Diagnostic Accuracy (STARD) guideline (S1 Checklist)." When completing the checklist, please use section and paragraph numbers, rather than page numbers.

* Please structure your Abstract according to STARD for Abstracts (https://www.equator-network.org/reporting-guidelines/stard-abstracts/).

* Please structure the Methods section using the following sub-headings: Study design, Participants, Test methods, Analysis.

* Please include a diagram to describe the flow of participants through the study (typically figure 1).

MENDELIAN RANDOMIZATION STUDIES

* Please ensure that the study is reported according to the STROBE-MR guideline (https://www.equator-network.org/reporting-guidelines/strobe/) and include the completed STROBE-MR checklist as Supporting Information. Please add the following statement, or similar, to the Methods: "This study is reported as per the Strengthening the Reporting of Observational Studies in Epidemiology (STROBE) guideline, specific for mendelian randomization (S1 Checklist)." When completing the checklist, please use section and paragraph numbers, rather than page numbers.

* In the Introduction, please describe the exposure and the evidence for a potential causal relationship between exposure and outcome.

* In the Methods, please explicitly state the 3 core instrumental variable assumptions for the main analysis (relevance, independence, and exclusion restriction), as well assumptions for any additional or sensitivity analysis.

* In the Methods, please describe the MR estimator (e.g., 2-stage least squares, Wald ratio) and related statistics. Detail the included covariates and, in case of 2-sample MR, whether the same covariate set was used for adjustment in the 2 samples.

* If you are presenting an instrumental variable estimate, please compare this to the conventional observational estimate.

* Report the associations between genetic variant and exposure and between genetic variant and outcome, preferably on an interpretable scale.

* Report MR estimates of the relationship between exposure and outcome and the measures of uncertainty from the MR analysis, on an interpretable scale, such as odds ratio or relative risk per SD difference.

* If relevant, please consider translating estimates of relative risk into absolute risk for a meaningful time period.

* Please consider including plots to visualize results (e.g., forest plot, scatterplot of associations between genetic variants and outcome vs between genetic variants and exposure).

SURVEY-BASED STUDIES

* Please ensure that the study is reported according to the CROSS guideline (https://www.equator-network.org/reporting-guidelines/a-consensus-based-checklist-for-reporting-of-survey-studies-cross/) and include the completed CROSS checklist as Supporting Information. Please add the following statement, or similar, to the Methods: "This study is reported as per A Consensus-Based Checklist for Reporting of Survey Studies (CROSS) guideline (S1 Checklist)." When completing the checklist, please use section and paragraph numbers, rather than page numbers.

* Please report your survey response rates according to AAPOR recommendations (https://aapor.org/standards-and-ethics/best-practices/)

* Please define how the population surveyed was sampled.

* Please compare characteristics of respondents and nonrespondents if possible.

* If sequential waves of the survey were sent, please specify whether the characteristics of respondents changed over time or remained constant.

* Please include the survey response rate in the Abstract.

* Please include a copy of the survey in the supplementary files.

SYSTEMATIC REVIEWS & META-ANALYSES

* Please report your SR/MA according to the PRISMA guidelines provided at the EQUATOR site. http://www.equator-network.org/reporting-guidelines/prisma/. Please provide the completed PRISMA checklist as Supporting Information. When completing the checklist, please use section and paragraph numbers, rather than page numbers. Please add the following statement, or similar, to the Methods: "This study is reported as per the Preferred Reporting Items for Systematic Reviews and Meta-Analyses (PRISMA) guideline (S1 Checklist)."

* Abstract: Please report your abstract according to PRISMA for abstracts (https://doi.org/10.1371/journal.pmed.1001419) following the PLOS Medicine abstract structure (Background, Methods and Findings, Conclusions). Please ensure you provide dates of search, data sources, number of studies included, types of study designs included, eligibility criteria, and synthesis/appraisal methods.

* Please note that we expect searches to be updated to within 6 months of the time of submission.

QUALITATIVE STUDIES

* Please report your qualitative study according to the appropriate study design provided at (http://www.equator-network.org/?post_type=eq_guidelines&eq_guidelines_study_design=qualitative-research&eq_guidelines_clinical_specialty=0&eq_guidelines_report_section=0&s=) and provide the relevant completed checklist as a supplemental file. In the checklist, please include sufficient text excerpted from the manuscript to explain how you accomplished all applicable items. When completing checklists, please use section and paragraph numbers, rather than page numbers.

* We recommend that authors use the COREQ checklist, or other relevant checklists listed by the Equator Network, such as the SRQR, to ensure complete reporting (see: http://www.equator-network.org/?post_type=eq_guidelines&eq_guidelines_study_design=qualitative-research&eq_guidelines_clinical_specialty=0&eq_guidelines_report_section=0&s=). Please add the following statement, or similar, to the Methods: "This study is reported as per the Consolidated criteria for reporting qualitative research (COREQ): a 32-item checklist for interviews and focus groups (S1 Checklist)."

* In general, we expect qualitative studies to include the following: 1) defined objectives or research questions; 2) description of the sampling strategy, including rationale for the recruitment method, participant inclusion/exclusion criteria and the number of participants recruited; 3) detailed reporting of the data collection procedures; 4) data analysis procedures described in sufficient detail to enable replication; 5) a discussion of potential sources of bias; and 6) a discussion of limitations.

HEALTH ECONOMICS / COST-EFFECTIVENESS STUDIES

* Please ensure that the study is reported according to the CHEERS guideline (available from: https://www.equator-network.org/reporting-guidelines/cheers) and include the completed checklist as Supporting Information. Please add the following statement, or similar, to the Methods: "This study is reported as per the Strengthening the Consolidated Health Economic Evaluation Reporting Standards 2022 (CHEERS 2022) Statement (S1 Checklist)." When completing the checklist, please use section and paragraph numbers, rather than page numbers.

---

## [Decision Letter · Decision Letter 2]

10 Jul 2025

Dear Dr Schwalb,

Many thanks for submitting your manuscript "Estimating the global burden of viable Mycobacterium tuberculosis infection" (PMEDICINE-D-24-03899R2) to PLOS Medicine. The paper has been reviewed by subject experts and a statistician; their comments are included below and can also be accessed here: [LINK]

As you will see, the reviewers appreciate your efforts to revise the manuscript, however they continue to express some concerns, including about uncertainty in the estimates, that warrant addressing. After discussing the paper with the editorial team and an academic editor with relevant expertise, I'm pleased to invite you to revise the paper in response to the reviewers' comments. We plan to send the revised paper to some or all of the original reviewers, and we cannot provide any guarantees at this stage regarding publication.

Please note that we ask you to provide additional statements in the Abstract and Author Summary regarding the limitations in the methodology and the conclusions, and reflecting the comments of the reviewers.

We ask that you submit your revision by Jul 31 2025 11:59PM. However, if this deadline is not feasible, please contact me by email, and we can discuss a suitable alternative.

Don't hesitate to contact me directly with any questions (afarrell@plos.org).

Best regards,

Alison

Alison Farrell, Ph.D.

Senior Editor

PLOS Medicine

afarrell@plos.org

Comments from the academic editor:

[ADD SPECIFIC POINTS OF EMPHASIS (EG, RAISED IN EDITORIAL MEETING)]

Comments from the reviewers:

Reviewer #1: Many thanks to the authors for the time and efforts taken to address the previous comments. They have clearly sought to do a thorough job. I do have some remaining concerns / comments relating to the robustness of the results:

1. I am still quite nervous about the precision of the estimates provided. Notably, they have changed considerably in the revised version, compared to the original. For example, 289 million people, representing 3.7% (95%UI: 3.1-4.3) of the global population, are now estimated to harbour a viable Mtb infection. This was previously stated in R1 as between 387 (95%UI:347-441) and 606 (95%UI:549-670) million people, or between 4.9% (95%UI:4.4-5.6) and 7.7% (95%UI:6.9-8.5) of the global population. Notably, the 95% uncertainty intervals between the original and new estimates do not overlap.

I appreciate that the authors have re-run their analyses, including aiming to account for more uncertainty in clearance of infection estimates and correcting for issues with ageing terms and contact mixing matrices. But it is quite disconcerting that the main analysis findings can change beyond the widths of the presented uncertainty intervals, between revisions. To me, this perhaps reflects the underlying truth that there is a great deal of uncertainty that underlies the present analysis. My strong preference would be to embrace and tackle this uncertainty head on (see comment (3) below), rather than risking "burying it" deep in the methods.

2. Related, the authors state that "Estimates of both overall infection burden and recent infection were robust across scenarios (Table S6 and Table S7)." However, Tables S6 and S7 do not seem to present any alternative scenarios, unless I have missed these? They seem to show numbers and proportions of individuals with viable Mtb infection by region. If this is the case, why are the results now robust, when they previously varied depending on long-term self-clearance assumptions?

3. I appreciate that the authors have now tried to incorporate uncertainty in the self-clearance estimates (shown in Figure S4). As a non-mathematical modeler, I am unclear on how to interpret the approach and on how it influences the downstream analyses. Do the final modelled estimates of viable Mtb infection include the uncertainty bounds shown in the self-clearance calibration plot (Fig S4) - or do only the point estimates really get pulled through? I would find this much easier to interpret if the authors could provide current Mtb viability estimates where the self-clearance rates are systematically varied across a plausible range (I previously suggested 50-90% or similar at 1 year - with a trajectory thereafter that reflects this). Otherwise, the findings risk being fundamentally based on calibration to a previous natural history mathematical model (Horton et al) that was itself based on historic data, but without any supporting empirical data. This feels quite risky given that the final publication is likely to be highly cited. I believe that my suggestion to clearly show the impact of systematically varying the self-clearance assumption (for which we have very weak underlying data) on the final estimates will be much more meaningful and interpretable to the broad readership of PLOS Medicine.

Reviewer #2: I would like to thank the authors for their revisions to their paper (particularly the revised approach to estimating clearance rates that results in increased uncertainty) and thoughtful responses. I have a couple of remaining concerns about the revised manuscript that should be possible to address through further revisions:

1. Regarding the previous comment #13 in the response document - I appreciate the authors clarifying that no correlations were included in parameters across countries, years, or age groups. However, I am still concerned that the lack of correlations, particularly in ARI estimates across countries, results in the uncertainty in country-specific values being to a large extent "washed out" in regional and global estimates - because the analysis is much more likely to pair high ARI samples from some countries with low samples from others, rather than including high samples from all countries - when in reality it is possible that components of the ARI estimation process (such as the 2.9x adjustment factor) result in systematic over- or under-estimation. For example, comparing Table S9 to Table 1 we can see that there is much more uncertainty at a country level than at the regional level. I would request again that the authors consider at least a sensitivity analysis that induces correlations across countries, or some other method that makes it more likely to sample high (or low) ARI values across most countries in the same run. If I understand the methods correctly I don't think this would require rerunning the analysis - just combining results across countries differently.

2. I understand that a formal validation procedure for comparing indirect and direct ARI estimates presents a number of challenges. Could the authors at least include an appendix table that compares direct ARI estimates (e.g., perhaps those from the past 20 or so years from Table S1) with the indirect ARI estimate for a corresponding age-country-year? I understand that a unique strength of the indirect estimates is that they are readily available for all countries and years, but if they have another advantage over the direct estimates that is expected to result in differences between the direct and indirect estimates that would be helpful to explain.

3. Regarding the previous comment #15, I agree that progression is non-linear and disease states after progression can be dynamic, but couldn't an exit from viable Mtb infection resulting from progression be modeled that matches the estimates from Horton et al. 2023? In that paper, by year 1 and beyond, the prevalence of "active" TB exceeded that of "viable infection" - so it would seem that after an initially high clearance rate, the dynamics of disease progression would start to become an important input into the number of viable infections that remain. Furthermore, it appears that progression is included in the model used to calibrate the clearance rates - so then using those calibrated rates in a model without progression is somewhat problematic. The statement "Even if progression had been included in the model, it is unlikely that this would lead to an overestimation of infection" may be true, but it seems very speculative at this point and I am not convinced (but am open to convincing as I may have missed something). If not, perhaps a sensitivity analysis that includes some exit due to progression based on the Horton 2023 paper could be a good solution to determine if this is indeed the case. If that is not possible, I'd like to see some attempt by the authors to address this in some other way.

---

* Please upload any figures associated with your paper as individual TIF or EPS files with 300dpi resolution at resubmission; please read our figure guidelines for more information on our requirements: http://journals.plos.org/plosmedicine/s/figures. While revising your submission, please upload your figure files to the PACE digital diagnostic tool, https://pacev2.apexcovantage.com/. PACE helps ensure that figures meet PLOS requirements. To use PACE, you must first register as a user. Then, login and navigate to the UPLOAD tab, where you will find detailed instructions on how to use the tool. If you encounter any issues or have any questions when using PACE, please email us at PLOSMedicine@plos.org.

* [EDITOR: CHECK FINANCIAL DISCLOSURES, COI, DAS, AND ETHICS STATEMENTS AND INCLUDE ANY NECESSARY REQUESTS]

* Please ensure that the study is reported according to the [XXXX] guideline and include the completed [XXXX] checklist as Supporting Information. When completing the checklist, please use section and paragraph numbers, rather than page numbers. Please add the following statement, or similar, to the Methods: "This study is reported as per [XXXX] guideline (S1 Checklist)."

FIGURES AND TABLES

SUPPLEMENTARY MATERIAL

REFERENCES

[STUDY TYPE-SPECIFIC REQUESTS - DELETE SECTIONS AS NECESSARY]

RCTs [REFER TO RCT CHECKLIST AND MEETING NOTES FOR DETAILS TO ADD]

* PLOS Medicine requires that all trials be prospectively registered in one of registries recognized by WHO. Please ensure that study registration details are included in the Methods section.

* Please structure the Methods section using the following sub-headings: Study design and participants, Randomization and masking, Procedures, Outcomes, Statistical analysis.

* The following outcomes measures [ADD DETAILS AS NEEDED OR DELETE BULLET POINT] appear to differ between the submitted manuscript and the protocol [and/or trial registry]. Please clarify and explain all discrepancies between the paper and protocol. If the outcomes were not prespecified in the protocol, please define them in the Methods (Outcomes section) as post hoc and explain why they were added. Post-hoc comparisons should be presented as hypothesis generating rather than conclusive.

* Please ensure that all prespecified outcomes (primary, secondary, and exploratory) are listed in the Methods/Outcomes section and indicate whether there are outcomes that are not presented in the current report.

* Please specify the dates (Month Day, Year) during which study enrollment and follow up occurred.

* Please include absolute numbers wherever you report percentages; eg, n/N (%)

* Please present the safety data for the study including numbers of specific events and whether or not adverse events are thought to be related to treatment. AEs should be reported in the abstract, per CONSORT and CONSORT-Harms.

* Please complete the CONSORT checklist (https://www.equator-network.org/reporting-guidelines/consort/) and ensure that all components of CONSORT are present in the manuscript, including how randomization was performed, allocation concealment, blinding of intervention, definition of lost to follow-up, power statement. When completing the checklist, please use section and paragraph numbers, rather than page numbers.

* Please report your abstract according to CONSORT for abstracts, following the PLOS Medicine abstract structure (Background, Methods and Findings, Conclusions) https://www.equator-network.org/reporting-guidelines/consort-abstracts/

* If your trial had to undergo important modifications in response to extenuating circumstances, please complete the CONSERVE-CONSORT checklist and provide in your Supporting Information; (https://www.equator-network.org/reporting-guidelines/guidelines-for-reporting-trial-protocols-and-completed-trials-modified-due-to-the-covid-19-pandemic-and-other-extenuating-circumstances-the-conserve-2021-statement/). When completing the checklist, please use section and paragraph numbers, rather than page numbers.

* In keeping with our commitment to Open Science, please include the study protocol document and analysis plan (including any amendments) as Supporting Information to be published with the manuscript if accepted.

* Please note that PLOS Medicine requires prospective, public registration of a data sharing plan (as part of mandatory clinical trials registration) for all clinical trials that began enrollment on or after January 1, 2019, in accordance with ICMJE requirements.

OBSERVATIONAL STUDIES

* Abstract: Please include the study design, population and setting, number of participants, years during which the study took place (enrollment and follow up), length of follow up, and main outcome measures.

* Please ensure that the study is reported according to the STROBE (or appropriate STOBE extension) guideline (available from: https://www.equator-network.org/reporting-guidelines/strobe) and include the completed STROBE (or STROBE extension) checklist as Supporting Information. Please add the following statement, or similar, to the Methods: "This study is reported as per the Strengthening the Reporting of Observational Studies in Epidemiology (STROBE) guideline (S1 Checklist)." When completing the checklist, please use section and paragraph numbers, rather than page numbers.

* [FOR POPULATION HEALTH/REGISTRY STUDIES] Please ensure that the study is reported according to the RECORD guideline (available from https://www.record-statement.org) and include the completed checklist as Supporting Information. Please add the following statement, or similar, to the Methods: "This study is reported as per the Reporting of Studies Conducted using Observational Routinely-Collected Data (RECORD) guideline (S1 Checklist)." When completing the checklist, please use section and paragraph numbers, rather than page numbers.

* [FOR POPULATION HEALTH ESTIMATES] Please ensure that the study is reported according to the GATHER statement (available from https://www.equator-network.org/reporting-guidelines/gather-statement) and include the completed checklist as Supporting Information. Please add the following statement, or similar, to the Methods: "This study is reported as per the Guidelines for Accurate and Transparent Health Estimates Reporting (GATHER) statement (S1 Checklist)." When completing the checklist, please use section and paragraph numbers, rather than page numbers.

* [FOR MEDIATION ANALYSES] We recommend that the study is reported according to the AGReMA statement (https://agrema-statement.org/#:~:text=AGReMA%20is%20an%20evidence%2D%20and,randomised%20trials%20and%20observational%20studies) and include the completed checklist as Supporting Information. Please add the following statement, or similar, to the Methods: "This study is reported as per the Guideline for Reporting Mediation Analyses (AGReMA) statement (S1 Checklist)." When completing the checklist, please use section and paragraph numbers, rather than page numbers.

* For all observational studies, in the manuscript text, please indicate: (1) the specific hypotheses you intended to test, (2) the analytical methods by which you planned to test them, (3) the analyses you actually performed, and (4) when reported analyses differ from those that were planned, transparent explanations for differences that affect the reliability of the study's results. If a reported analysis was performed based on an interesting but unanticipated pattern in the data, please be clear that the analysis was data driven.

* Please state in the Methods section whether the study had a prospective protocol or analysis plan. If a prospective analysis plan (from your funding proposal, IRB or other ethics committee submission, study protocol, or other planning document written before analyzing the data) was used in designing the study, please include the relevant document(s) with your revised manuscript as a Supporting Information file to be published alongside your study and cite it in the Methods section. A legend for this file should be included at the end of your manuscript. If no such document exists, please make sure that the Methods section transparently describes when analyses were planned, and when/why any data-driven changes to analyses took place. Changes in the analysis, including those made in response to peer review comments, should be identified as such in the Methods section of the paper, with rationale.

MODELLING STUDIES

The following list is derived from Geoffrey P Garnett, Simon Cousens, Timothy B Hallett, Richard Steketee, Neff Walker. Mathematical models in the evaluation of health programmes. (2011) Lancet DOI:10.1016/S0140-6736(10)61505-X:

* If pertinent, please provide a diagram that shows the model structure, including how the natural history of the disease is represented, the process and determinants of disease acquisition, and how the putative intervention could affect the system.

* Please provide a complete list of model parameters, including clear and precise descriptions of the meaning of each parameter, together with the values or ranges for each, with justification or the primary source cited and important caveats about the use of these values noted.

* Please provide a clear statement about how the model was fitted to the data, including goodness-of-fit measure, the numerical algorithm used, which parameter varied, constraints imposed on parameter values, and starting conditions.

* For uncertainty analyses, please state the sources of uncertainties quantified and not quantified [can include parameter, data, and model structure].

* Please provide sensitivity analyses to identify which parameter values are most important in the model. Uncertainty estimates seek to derive a range of credible results on the basis of an exploration of the range of reasonable parameter values. The choice of method should be presented and justified.

* Please discuss the scientific rationale for the choice of model structure and identify points where this choice could influence conclusions drawn. Please also describe the strength of the scientific basis underlying the key model assumptions.

* For studies that develop a prediction model or evaluate its performance, please ensure that the study is reported according to the TRIPOD statement (https://www.equator-network.org/reporting-guidelines/tripod-statement) and include the completed checklist as Supporting Information. Please add the following statement, or similar, to the Methods: "This study is reported as per the Transparent Reporting of a Multivariable Prediction Model for Individual Prognosis Or Diagnosis (TRIPOD) statement (S1 Checklist)." For studies using machine learning, please use the TRIPOD-AI checklist. When completing the checklist, please use section and paragraph numbers, rather than page numbers.

DIAGNOSTIC STUDIES

* Please ensure that the study is reported according to the STARD guideline (https://www.equator-network.org/reporting-guidelines/stard/) and include the completed STARD checklist as Supporting Information. Please add the following statement, or similar, to the Methods: "This study is reported as per the Standards for Reporting of Diagnostic Accuracy (STARD) guideline (S1 Checklist)." When completing the checklist, please use section and paragraph numbers, rather than page numbers.

* Please structure your Abstract according to STARD for Abstracts (https://www.equator-network.org/reporting-guidelines/stard-abstracts/).

* Please structure the Methods section using the following sub-headings: Study design, Participants, Test methods, Analysis.

* Please include a diagram to describe the flow of participants through the study (typically figure 1).

MENDELIAN RANDOMIZATION STUDIES

* Please ensure that the study is reported according to the STROBE-MR guideline (https://www.equator-network.org/reporting-guidelines/strobe/) and include the completed STROBE-MR checklist as Supporting Information. Please add the following statement, or similar, to the Methods: "This study is reported as per the Strengthening the Reporting of Observational Studies in Epidemiology (STROBE) guideline, specific for mendelian randomization (S1 Checklist)." When completing the checklist, please use section and paragraph numbers, rather than page numbers.

* In the Introduction, please describe the exposure and the evidence for a potential causal relationship between exposure and outcome.

* In the Methods, please explicitly state the 3 core instrumental variable assumptions for the main analysis (relevance, independence, and exclusion restriction), as well assumptions for any additional or sensitivity analysis.

* In the Methods, please describe the MR estimator (e.g., 2-stage least squares, Wald ratio) and related statistics. Detail the included covariates and, in case of 2-sample MR, whether the same covariate set was used for adjustment in the 2 samples.

* If you are presenting an instrumental variable estimate, please compare this to the conventional observational estimate.

* Report the associations between genetic variant and exposure and between genetic variant and outcome, preferably on an interpretable scale.

* Report MR estimates of the relationship between exposure and outcome and the measures of uncertainty from the MR analysis, on an interpretable scale, such as odds ratio or relative risk per SD difference.

* If relevant, please consider translating estimates of relative risk into absolute risk for a meaningful time period.

* Please consider including plots to visualize results (e.g., forest plot, scatterplot of associations between genetic variants and outcome vs between genetic variants and exposure).

SURVEY-BASED STUDIES

* Please ensure that the study is reported according to the CROSS guideline (https://www.equator-network.org/reporting-guidelines/a-consensus-based-checklist-for-reporting-of-survey-studies-cross/) and include the completed CROSS checklist as Supporting Information. Please add the following statement, or similar, to the Methods: "This study is reported as per A Consensus-Based Checklist for Reporting of Survey Studies (CROSS) guideline (S1 Checklist)." When completing the checklist, please use section and paragraph numbers, rather than page numbers.

* Please report your survey response rates according to AAPOR recommendations (https://aapor.org/standards-and-ethics/best-practices/)

* Please define how the population surveyed was sampled.

* Please compare characteristics of respondents and nonrespondents if possible.

* If sequential waves of the survey were sent, please specify whether the characteristics of respondents changed over time or remained constant.

* Please include the survey response rate in the Abstract.

* Please include a copy of the survey in the supplementary files.

SYSTEMATIC REVIEWS & META-ANALYSES

* Please report your SR/MA according to the PRISMA guidelines provided at the EQUATOR site. http://www.equator-network.org/reporting-guidelines/prisma/. Please provide the completed PRISMA checklist as Supporting Information. When completing the checklist, please use section and paragraph numbers, rather than page numbers. Please add the following statement, or similar, to the Methods: "This study is reported as per the Preferred Reporting Items for Systematic Reviews and Meta-Analyses (PRISMA) guideline (S1 Checklist)."

* Abstract: Please report your abstract according to PRISMA for abstracts (https://doi.org/10.1371/journal.pmed.1001419) following the PLOS Medicine abstract structure (Background, Methods and Findings, Conclusions). Please ensure you provide dates of search, data sources, number of studies included, types of study designs included, eligibility criteria, and synthesis/appraisal methods.

* Please note that we expect searches to be updated to within 6 months of the time of submission.

QUALITATIVE STUDIES

* Please report your qualitative study according to the appropriate study design provided at (http://www.equator-network.org/?post_type=eq_guidelines&eq_guidelines_study_design=qualitative-research&eq_guidelines_clinical_specialty=0&eq_guidelines_report_section=0&s=) and provide the relevant completed checklist as a supplemental file. In the checklist, please include sufficient text excerpted from the manuscript to explain how you accomplished all applicable items. When completing checklists, please use section and paragraph numbers, rather than page numbers.

* We recommend that authors use the COREQ checklist, or other relevant checklists listed by the Equator Network, such as the SRQR, to ensure complete reporting (see: http://www.equator-network.org/?post_type=eq_guidelines&eq_guidelines_study_design=qualitative-research&eq_guidelines_clinical_specialty=0&eq_guidelines_report_section=0&s=). Please add the following statement, or similar, to the Methods: "This study is reported as per the Consolidated criteria for reporting qualitative research (COREQ): a 32-item checklist for interviews and focus groups (S1 Checklist)."

* In general, we expect qualitative studies to include the following: 1) defined objectives or research questions; 2) description of the sampling strategy, including rationale for the recruitment method, participant inclusion/exclusion criteria and the number of participants recruited; 3) detailed reporting of the data collection procedures; 4) data analysis procedures described in sufficient detail to enable replication; 5) a discussion of potential sources of bias; and 6) a discussion of limitations.

HEALTH ECONOMICS / COST-EFFECTIVENESS STUDIES

* Please ensure that the study is reported according to the CHEERS guideline (available from: https://www.equator-network.org/reporting-guidelines/cheers) and include the completed checklist as Supporting Information. Please add the following statement, or similar, to the Methods: "This study is reported as per the Strengthening the Consolidated Health Economic Evaluation Reporting Standards 2022 (CHEERS 2022) Statement (S1 Checklist)." When completing the checklist, please use section and paragraph numbers, rather than page numbers.

---

## [Decision Letter · Decision Letter 3]

13 Nov 2025

Dear Dr Schwalb,

Many thanks for submitting your manuscript "Estimating the global burden of viable Mycobacterium tuberculosis infection" (PMEDICINE-D-24-03899R3) to PLOS Medicine. The revised paper has been reviewed by subject experts and a statistician (reviewer 3). I apologize for the delay in conveying to you our decision as we were waiting for final comments from the reviewers. Their comments are included below and can also be accessed here: [LINK]

As you will see, reviewers 1 and 2 have a few remaining concerns that should be addressed in a revised manuscript. In particular, the reviewers require a more explicit discussion of the study limitations and in particular on the uncertainty surrounding self-clearance rates. After discussing the paper with the editorial team and an academic editor with relevant expertise, I'm pleased to invite you to revise the paper in response to the reviewers' comments. In particular, we ask you to acknowledge the reviewers' concerns in the manuscript, provide a response to the comments of reviewer 2 and incorporate such a response in the manuscript for transparency and assessment by readers. Please revise the Abstract and Author Summary accordingly, and add a paragraph in the Discussion to advise the field on how to interpret these data and offer suggestions on how to validate the conclusions. The limitations of the study should be clearly articulated throughout. Please be advised that we may discuss a revised manuscript with the reviewers or academic editor and we cannot provide any guarantees at this stage regarding publication.

We ask that you submit your revision by Dec 04 2025 11:59PM. However, if this deadline is not feasible, please contact me by email, and we can discuss a suitable alternative.

Don't hesitate to contact me directly with any questions (afarrell@plos.org).

Best regards,

Alison

Alison Farrell, Ph.D.

Senior Editor

PLOS Medicine

afarrell@plos.org

Comments from the reviewers:

Reviewer #1: Thank you to the authors for their further revisions. The additional sensitivity analyses (table S13) are very helpful. I have a few more suggestions relating to the self-clearance assumptions / sensitivity analyses:

1. The new finding that "global burden estimates were particularly sensitive to lower self-clearance rates, highlighting the need for better empirical data to inform this parameter" is quite critical and is a key take-home message from this manuscript, since the evidence underpinning the primary self-clearance estimates are so weak. I would strongly suggest that these findings are included in both the abstract and author summary.

2. Related, I would also strongly recommend adding a figure that visually represents the findings from table S13 in the main manuscript.

3. Please can the authors clarify what the self-clearance rate variation of +-25-75% in the sensitivity analyses actually means? For example, how can a baseline assumption of 80.9% self-clearance at 1 year increase by 75% in the sens analysis?

4. The limitations section should explicitly acknowledge the weakness of the underlying data on self-clearance, along with the major impact that this has on the results.

5. The abstract states that "Self-clearance rates were informed by empirical data and modelling estimates". But is this really true, given that only the Horton et al modelling estimates are explained and cited as source data for the self-clearance rates in the methods?

Reviewer #2: I appreciate the authors' consideration of my previous comments and would like to thank them for adding ARI estimates to the appendix and for clarifying some points about the methods. I believe the authors are doing their best to be responsive to the reviewers' comments, but I still have some concerns about this analysis. A couple of additional comments for the authors' consideration, based on those responses, are:

1. I share Reviewer 1's apprehension about the handling of uncertainty in this analysis. My specific concerns relate more to the ARI estimation process, including no correlation between the many elements of the ARI estimation process that do not represent meaningful cross-country variation (such as the duration of disease, 2.9x adjustment, and revised Styblo rule) resulting in underestimation of uncertainty at the regional and global level (I was not suggesting that the authors induce 100% correlation between these parameters). I am happy to see the ARI estimates shown in more detail in the appendix but am concerned about the extent to which direct and indirect estimates do not match (in almost half of the country-years where direct estimates are available, the confidence intervals between direct and indirect do not overlap). While I appreciate the sensitivity analysis on clearance rates, I would like to see some attempt to better address uncertainty in ARI (my suggestion was to induce a correlation between country estimates that could be < 100%, say an arbitrary 50%, but this could be handled differently) or to acknowledge these uncertainties much more in the limitations section.

2. The fact that, in this model, viable infection includes active disease introduces some additional issues that were not apparent previously. The main issue is that, in reality, those with active disease can "clear" their infections (i.e., recover from TB disease) by being treated, not just through natural recovery following disease. In the model, these individuals should therefore face high rates of exit out of "viable infection" - but if I understand correctly this is not being captured because the clearance rates are calibrated to output from Horton et al. 2023, which did not include treatment. Omitting exits due to treatment of active disease is therefore likely to result in an overestimation of the prevalence of viable infection and of recent infection.

There are some additional concerns that probably have less influence on the overall numbers (although it's difficult to say without any sensitivity analysis on this) and so I will emphasize them less. But they include (1) that people with TB disease should face higher mortality rates. The supplementary materials statement "Note that when formulating dynamics in terms of age-stratified proportions in a dynamic population, age-specific mortality rates cancel out" would seem to no longer apply if mortality rates vary by time since infection, which again they may if the proportion of "clearance" that is due to progression and recovery (vs. infection clearance without ever progressing) is higher during earlier years of infection. And (2) the model allows for reinfection, with some protective factor, which doesn't really make sense for people with active disease. While it's true that people can be co-infected (or "co-diseased") with multiple Mtb strains, in the model this would send these individuals back to "infection year 1". Again, I suspect that these 2 limitations have little influence on the overall numbers generated by this analysis, but the magnitude of overestimation from excluding clearance/recovery due to treatment is less clear, without any sensitivity analyses being run. I agree with the authors that such an analysis would require a more complex model, but perhaps given the limitations of the current approach, a more complex model is warranted.

Without any sensitivity analysis on this, my suggestion is that much of the manuscript framing and wording seem to need revision if the estimates of viable infection include people with prevalent/active TB disease. Referring to the rate of transition from infection to uninfected as a "self-clearance" rate seems inaccurate. As another example, the paper repeatedly alludes to the utility of these estimates in guiding TPT provision, but people who have active TB disease should not be given TPT. As I understand it, we cannot just adjust for the estimates of recent viable infection by subtracting out estimates of active TB prevalence, because we do not know the proportion of active TB prevalence that is from recent infection. While active disease is likely to represent a minority of infection prevalence, this caveat limits the extent to which the numbers presented are "medically-actionable". I would like to see the framing of the manuscript and some of the model terminology revisited to ensure it is appropriately describing what is actually being estimated.

Reviewer #3: The authors have revised their paper and this is now a substantially improved manuscript, acceptable for publication.

---

Please include a response to these requirements in your resubmission.

* Please upload any figures associated with your paper as individual TIF or EPS files with 300dpi resolution at resubmission; please read our figure guidelines for more information on our requirements: http://journals.plos.org/plosmedicine/s/figures. While revising your submission, we strongly recommend that you use PLOS's NAAS tool (https://ngplosjournals.pagemajik.ai/artanalysis) to test your figure files. NAAS can convert your figure files to the TIFF file type and meet basic requirements (such as print size, resolution), or provide you with a report on issues that do not meet our requirements and that NAAS cannot fix.

After uploading your figures to PLOS's NAAS tool - https://ngplosjournals.pagemajik.ai/artanalysis, NAAS will process the files provided and display the results in the "Uploaded Files" section of the page as the processing is complete.

If the uploaded figures meet our requirements (or NAAS is able to fix the files to meet our requirements), the figure will be marked as "fixed" above. If NAAS is unable to fix the files, a red "failed" label will appear above.

When NAAS has confirmed that the figure files meet our requirements, please download the file via the download option, and include these NAAS processed figure files when submitting your revised manuscript.

*FORMATTING - GENERAL

*** Please confirm that your title complies with PLOS Medicine's style. Your title must be nondeclarative and not a question. It should begin with main concept if possible. "Effect of" should be used only if causality can be inferred, i.e., for an RCT. Please place the study design ("A randomized controlled trial," "A retrospective study," "A modelling study," etc.) in the subtitle (ie, after a colon).

** In the last sentence of the Abstract Methods and Findings section, please describe the main limitation(s) of the study's methodology.

** In the final bullet point of 'What Do These Findings Mean?', in the Author Summary, please include the main limitations of the study in non-technical language. Please see our author guidelines for more information: https://journals.plos.org/plosmedicine/s/revising-your-manuscript#loc-author-summary.

* Please include the URL for the funder.

* Please confirm that the appropriate usage rights apply to the use of maps in the figures. Please see our guidelines for map images: https://journals.plos.org/plosmedicine/s/figures#loc-maps

* Please use commas rather than hyphens in confidence intervals.

FIGURES AND TABLES

SUPPLEMENTARY MATERIAL

REFERENCES

MODELLING STUDIES

The following list is derived from Geoffrey P Garnett, Simon Cousens, Timothy B Hallett, Richard Steketee, Neff Walker. Mathematical models in the evaluation of health programmes. (2011) Lancet DOI:10.1016/S0140-6736(10)61505-X:

* If pertinent, please provide a diagram that shows the model structure, including how the natural history of the disease is represented, the process and determinants of disease acquisition, and how the putative intervention could affect the system.

* Please provide a complete list of model parameters, including clear and precise descriptions of the meaning of each parameter, together with the values or ranges for each, with justification or the primary source cited and important caveats about the use of these values noted.

* Please provide a clear statement about how the model was fitted to the data, including goodness-of-fit measure, the numerical algorithm used, which parameter varied, constraints imposed on parameter values, and starting conditions.

* For uncertainty analyses, please state the sources of uncertainties quantified and not quantified [can include parameter, data, and model structure].

* Please provide sensitivity analyses to identify which parameter values are most important in the model. Uncertainty estimates seek to derive a range of credible results on the basis of an exploration of the range of reasonable parameter values. The choice of method should be presented and justified.

* Please discuss the scientific rationale for the choice of model structure and identify points where this choice could influence conclusions drawn. Please also describe the strength of the scientific basis underlying the key model assumptions.

* For studies that develop a prediction model or evaluate its performance, please ensure that the study is reported according to the TRIPOD statement (https://www.equator-network.org/reporting-guidelines/tripod-statement) and include the completed checklist as Supporting Information. Please add the following statement, or similar, to the Methods: "This study is reported as per the Transparent Reporting of a Multivariable Prediction Model for Individual Prognosis Or Diagnosis (TRIPOD) statement (S1 Checklist)." For studies using machine learning, please use the TRIPOD-AI checklist. When completing the checklist, please use section and paragraph numbers, rather than page numbers.

DIAGNOSTIC STUDIES

* Please ensure that the study is reported according to the STARD guideline (https://www.equator-network.org/reporting-guidelines/stard/) and include the completed STARD checklist as Supporting Information. Please add the following statement, or similar, to the Methods: "This study is reported as per the Standards for Reporting of Diagnostic Accuracy (STARD) guideline (S1 Checklist)." When completing the checklist, please use section and paragraph numbers, rather than page numbers.

* Please structure your Abstract according to STARD for Abstracts (https://www.equator-network.org/reporting-guidelines/stard-abstracts/).

* Please structure the Methods section using the following sub-headings: Study design, Participants, Test methods, Analysis.

* Please include a diagram to describe the flow of participants through the study (typically figure 1).

MENDELIAN RANDOMIZATION STUDIES

* Please ensure that the study is reported according to the STROBE-MR guideline (https://www.equator-network.org/reporting-guidelines/strobe/) and include the completed STROBE-MR checklist as Supporting Information. Please add the following statement, or similar, to the Methods: "This study is reported as per the Strengthening the Reporting of Observational Studies in Epidemiology (STROBE) guideline, specific for mendelian randomization (S1 Checklist)." When completing the checklist, please use section and paragraph numbers, rather than page numbers.

* In the Introduction, please describe the exposure and the evidence for a potential causal relationship between exposure and outcome.

* In the Methods, please explicitly state the 3 core instrumental variable assumptions for the main analysis (relevance, independence, and exclusion restriction), as well assumptions for any additional or sensitivity analysis.

* In the Methods, please describe the MR estimator (e.g., 2-stage least squares, Wald ratio) and related statistics. Detail the included covariates and, in case of 2-sample MR, whether the same covariate set was used for adjustment in the 2 samples.

* If you are presenting an instrumental variable estimate, please compare this to the conventional observational estimate.

* Report the associations between genetic variant and exposure and between genetic variant and outcome, preferably on an interpretable scale.

* Report MR estimates of the relationship between exposure and outcome and the measures of uncertainty from the MR analysis, on an interpretable scale, such as odds ratio or relative risk per SD difference.

* If relevant, please consider translating estimates of relative risk into absolute risk for a meaningful time period.

* Please consider including plots to visualize results (e.g., forest plot, scatterplot of associations between genetic variants and outcome vs between genetic variants and exposure).

SURVEY-BASED STUDIES

* Please ensure that the study is reported according to the CROSS guideline (https://www.equator-network.org/reporting-guidelines/a-consensus-based-checklist-for-reporting-of-survey-studies-cross/) and include the completed CROSS checklist as Supporting Information. Please add the following statement, or similar, to the Methods: "This study is reported as per A Consensus-Based Checklist for Reporting of Survey Studies (CROSS) guideline (S1 Checklist)." When completing the checklist, please use section and paragraph numbers, rather than page numbers.

* Please report your survey response rates according to AAPOR recommendations (https://aapor.org/standards-and-ethics/best-practices/)

* Please define how the population surveyed was sampled.

* Please compare characteristics of respondents and nonrespondents if possible.

* If sequential waves of the survey were sent, please specify whether the characteristics of respondents changed over time or remained constant.

* Please include the survey response rate in the Abstract.

* Please include a copy of the survey in the supplementary files.

SYSTEMATIC REVIEWS & META-ANALYSES

* Please report your SR/MA according to the PRISMA guidelines provided at the EQUATOR site. http://www.equator-network.org/reporting-guidelines/prisma/. Please provide the completed PRISMA checklist as Supporting Information. When completing the checklist, please use section and paragraph numbers, rather than page numbers. Please add the following statement, or similar, to the Methods: "This study is reported as per the Preferred Reporting Items for Systematic Reviews and Meta-Analyses (PRISMA) guideline (S1 Checklist)."

* Abstract: Please report your abstract according to PRISMA for abstracts (https://doi.org/10.1371/journal.pmed.1001419) following the PLOS Medicine abstract structure (Background, Methods and Findings, Conclusions). Please ensure you provide dates of search, data sources, number of studies included, types of study designs included, eligibility criteria, and synthesis/appraisal methods.

* Please note that we expect searches to be updated to within 6 months of the time of submission.

QUALITATIVE STUDIES

* Please report your qualitative study according to the appropriate study design provided at (http://www.equator-network.org/?post_type=eq_guidelines&eq_guidelines_study_design=qualitative-research&eq_guidelines_clinical_specialty=0&eq_guidelines_report_section=0&s=) and provide the relevant completed checklist as a supplemental file. In the checklist, please include sufficient text excerpted from the manuscript to explain how you accomplished all applicable items. When completing checklists, please use section and paragraph numbers, rather than page numbers.

* We recommend that authors use the COREQ checklist, or other relevant checklists listed by the Equator Network, such as the SRQR, to ensure complete reporting (see: http://www.equator-network.org/?post_type=eq_guidelines&eq_guidelines_study_design=qualitative-research&eq_guidelines_clinical_specialty=0&eq_guidelines_report_section=0&s=). Please add the following statement, or similar, to the Methods: "This study is reported as per the Consolidated criteria for reporting qualitative research (COREQ): a 32-item checklist for interviews and focus groups (S1 Checklist)."

* In general, we expect qualitative studies to include the following: 1) defined objectives or research questions; 2) description of the sampling strategy, including rationale for the recruitment method, participant inclusion/exclusion criteria and the number of participants recruited; 3) detailed reporting of the data collection procedures; 4) data analysis procedures described in sufficient detail to enable replication; 5) a discussion of potential sources of bias; and 6) a discussion of limitations.

HEALTH ECONOMICS / COST-EFFECTIVENESS STUDIES

* Please ensure that the study is reported according to the CHEERS guideline (available from: https://www.equator-network.org/reporting-guidelines/cheers) and include the completed checklist as Supporting Information. Please add the following statement, or similar, to the Methods: "This study is reported as per the Strengthening the Consolidated Health Economic Evaluation Reporting Standards 2022 (CHEERS 2022) Statement (S1 Checklist)." When completing the checklist, please use section and paragraph numbers, rather than page numbers.

---

## [Decision Letter · Decision Letter 4]

8 Jan 2026

Dear Dr. Schwalb,

Thank you very much for re-submitting your manuscript "Estimating the global burden of viable Mycobacterium tuberculosis infection" (PMEDICINE-D-24-03899R4) for review by PLOS Medicine.

I have discussed the paper with my colleagues and the academic editor and it was also seen again by one reviewer. I am pleased to say that provided the remaining editorial and production issues are dealt with we are planning to accept the paper for publication in the journal.

The remaining issues that need to be addressed are listed at the end of this email. You will note that the reviewer requests that you clearly acknowledge in the Abstract and Author Summary the uncertainty of the estimates. Any accompanying reviewer attachments can be seen via the link below. Please take these into account before resubmitting your manuscript:

[LINK]

We look forward to receiving the revised manuscript by Jan 15 2026 11:59PM.

Sincerely,

Alison Farrell, Ph.D.

Senior Editor

PLOS Medicine

plosmedicine.org

Requests from Editors:

* Please review your text for claims of novelty or primacy (e.g. 'for the first time') and remove this language. In addition, please check that any use of statistical terms (such as trend or significant) are supported by the data, and if not please remove them.

* In the author summary, in the final bullet point of 'What Do These Findings Mean?', please include the main limitations of the study in non-technical language.

* Please add this statement to the manuscript's Competing Interests: "PJD is a paid statistical consultant on PLOS Medicine's statistical board."

* Please convert any stacked bar charts to another data representation for example a table, or other type of graph.

* Please confirm that the appropriate usage rights apply to the use of this map. Please see our guidelines for map images: https://journals.plos.org/plosmedicine/s/figures#loc-maps

*In the final bullet point of ‘What Do These Findings Mean?’ Please include the main limitations of the study in non-technical language.

Please see our author guidelines for more information: https://journals.plos.org/plosmedicine/s/revising-your-manuscript#loc-author-summary.

* Please confirm that your title complies with PLOS Medicine's style. Your title must be nondeclarative and not a question. It should begin with main concept if possible. "Effect of" should be used only if causality can be inferred, i.e., for an RCT. Please place the study design ("A randomized controlled trial," "A retrospective study," "A modelling study," etc.) in the subtitle (ie, after a colon). The current title does not adhere to our style.

* Please confirm that your abstract complies with our requirements, including format (three sections: Background, Methods and Findings, and Conclusions) and providing all the information relevant to this study type https://journals.plos.org/plosmedicine/s/submission-guidelines#loc-abstract

* Please ensure that the Introduction ends with a clear description of the study question or hypothesis.

* Please ensure that all abbreviations are defined at first use throughout the text.

* Please confirm that all numbers presented in the abstract are present and identical to numbers presented in the main manuscript text.

FUNDING STATEMENT

* The funding statement should include: specific grant numbers, initials of authors who received each award, URLs to sponsors’ websites. Also, please state whether any sponsors or funders (other than the named authors) played any role in study design, data collection and analysis, the decision to publish, or preparation of the manuscript. If they had no role in the research, include this sentence: “The funders had no role in study design, data collection and analysis, decision to publish, or preparation of the manuscript.”

The funders' URLs are missing from the statement.

For modeling studies:

The points below are derived from Geoffrey P Garnett, Simon Cousens, Timothy B Hallett, Richard Steketee, Neff Walker. Mathematical models in the evaluation of health programmes. (2011) Lancet DOI:10.1016/S0140-6736(10)61505-X:

* Please provide a diagram that shows the model structure, including how the disease natural history is represented, the process and determinants of disease acquisition, and how the putative intervention could affect the system.

* Please provide a complete list of model parameters, including clear and precise descriptions of the meaning of each parameter, together with the values or ranges for each, with justification or the primary source cited, and important caveats about the use of these values noted.

* Please provide a clear statement about how the model was fitted to the data including where relevant goodness-of-fit measure, the numerical algorithm used, which parameter varied, constraints imposed on parameter values, and starting conditions.

* For uncertainty analyses, please state the sources of uncertainties quantified and not quantified this can include parameter, data, and model structure.

* Please provide sensitivity analyses to identify which parameter values are most important in the model. Uncertainty estimates seek to derive a range of credible results on the basis of an exploration of the range of reasonable parameter values. The choice of method should be presented and justified.

* Please discuss the scientific rationale for this choice of model structure and identify points where this choice could influence conclusions drawn. Please also describe the strength of the scientific basis underlying the key model assumptions.

Comments from Reviewers:

Reviewer #1: Thank you to the authors for the revisions; the manuscript has been further improved. Some minor (final!) suggestions:

Given the considerable uncertainty, please highlight this in the final sections of the abstract and author summary. e.g:

Abstract Conclusions: "Our findings offer the first global burden estimates of viable Mtb infection, albeit with considerable uncertainty owing to weak underlying data."

Author Summary What do these findings mean?: "Future estimates would benefit from more contemporary, empirically derived annual risk of infection data through repeated immunoreactivity surveys, along with empirical measures of self-clearance."

[LINK]

---

## [Editor Report · Decision Letter 5]

20 Jan 2026

Dear Dr Schwalb,

On behalf of my colleagues and the Academic Editor, Madhukar Pai, I am pleased to inform you that we have agreed to publish your manuscript "Estimating the global burden of viable Mycobacterium tuberculosis infection: A mathematical modelling study" (PMEDICINE-D-24-03899R5) in PLOS Medicine.

PRESS

Sincerely,

Alison Farrell, Ph.D.

Senior Editor

PLOS Medicine